EMBO
Molecular Medicine

# Generalizable anchor aptamer strategy for loading nucleic acid therapeutics on exosomes

Gang Han[1], Yao Zhang[1], Li Zhong[1], Biaobiao Wang[1], Shuai Qiu[1], Jun Song[1], Caorui Lin [1], Fangdi Zou[2], Jingqiao Wu[1], Huanan Yu[1], Chao Liang[3], Ke Wen [4], Yiqi Seow [5] & HaiFang Yin [1,6] ✉

## Abstract

**Clinical deployment of oligonucleotides requires delivery technologies that improve stability, target tissue accumulation and cellular internalization. Exosomes show potential as ideal delivery vehicles. However, an affordable generalizable system for efficient loading of oligonucleotides on exosomes remain lacking. Here, we identified an Exosomal Anchor DNA Aptamer (EAA) via SELEX against exosomes immobilized with our proprietary CP05 peptides. EAA shows high binding affinity to different exosomes and enables efficient loading of nucleic acid drugs on exosomes. Serum stability of thrombin inhibitor NU172 was prolonged by exosome-loading, resulting in increased blood flow after injury in vivo. Importantly, Duchenne Muscular Dystrophy PMO can be readily loaded on exosomes via EAA (EXO$_{EAA-PMO}$). EXO$_{EAA-PMO}$ elicited significantly greater muscle cell uptake, tissue accumulation and dystrophin expression than PMO in vitro and in vivo. Systemic administration of EXO$_{EAA-PMO}$ elicited therapeutic levels of dystrophin restoration and functional improvements in *mdx* mice. Altogether, our study demonstrates that EAA enables efficient loading of different nucleic acid drugs on exosomes, thus providing an easy and generalizable strategy for loading nucleic acid therapeutics on exosomes.**

**Keywords** Exosome; Nucleic Acid Therapeutics; Anchor Aptamer; Drug Delivery; SELEX
**Subject Categories** Biotechnology & Synthetic Biology; Methods & Resources; Pharmacology & Drug Discovery

## Introduction

Oligonucleotides as a novel therapeutic modality have expanded rapidly in the past two decade, with more than a dozen of oligonucleotide drugs approved clinically and over a hundred in late-stage clinical development (Kulkarni et al, 2021; Roberts et al, 2020). However, efficient delivery of oligonucleotides to target tissues is key to unleashing their full clinical potential. A myriad of viral and nonviral delivery technologies have been developed to improve serum stability, cellular internalization, and target tissue accumulation including adenovirus-associated virus, adenovirus, lipid nanoparticles, cell-penetrating/tissue-targeting peptides, polymers, and other delivery tools (Kulkarni et al, 2021; Merdan et al, 2002; Paunovska et al, 2022; Roberts et al, 2020), each with its own pros and cons.

Nanosized intercellular messenger exosomes, secreted by most mammalian cells, show promise as natural biological delivery vehicles for nucleic acid drugs due to their good biocompatibility, low immunogenicity, and long circulation time (Gao et al, 2018; Herrmann et al, 2021; Wu et al, 2021). Exogenous oligonucleotides are primarily loaded onto exosomes by encapsulation and surface modification (He et al, 2018; Herrmann et al, 2021; Wu et al, 2021). Encapsulation refers to electroporation, sonication, extrusion or transfection, frequently with low loading efficiencies (He et al, 2018; Pi et al, 2018). Direct surface modification with nucleic acid drugs is challenging, so far realized only via membrane phospholipid bilayer insertion of cholesterol or exosomal anchor peptide (CP05) with variable efficiencies (Gao et al, 2018; Haraszti et al, 2018; Kamerkar et al, 2022; Pi et al, 2018). Using nucleic acid aptamers for surface modification can offer unique advantages as oligonucleotides of different chemistries can be directly synthesized with or annealed to aptamers and thus can be manufactured more affordably. Although the DNA aptamer targeting CD63, a protein biomarker of exosomes (Thery et al, 2018), has been extensively exploited for the purpose of exosome capture or detection in various forms (Wu et al, 2021; Zhu et al, 2020), it has not been used to load nucleic acid drugs onto exosomes. Moreover, as we are keen to develop orthogonal methods of surface modifications so as to combine peptide targeting with oligonucleotide loading, panning aptamers against CD63 can interfere with our CD63-targeting peptide CP05 binding on the same exosome.

Here, we screened a single-stranded DNA aptamer library via the systematic evolution of ligands by exponential enrichment (SELEX)

[1]State Key Laboratory of Experimental Hematology & The Province and Ministry Co-sponsored Collaborative Innovation Center for Medical Epigenetics & International Joint Laboratory of Ocular Diseases (Ministry of Education), School of Medical Technology & School of Basic Medical Sciences, Tianjin Medical University, Qixiangtai Road, Heping District, 300070 Tianjin, China. [2]Public Laboratory & Key Laboratory of Cancer Prevention and Therapy, Tianjin Medical University Cancer Institute and Hospital, National Clinical Research Center & Tianjin's Clinical Research Center for Cancer, 300060 Tianjin, China. [3]Department of Systems Biology, School of Life Sciences, Southern University of Science and Technology, 518055 Shenzhen, China. [4]Department of Pharmacology, Tianjin Key Laboratory of Inflammatory Biology, School of Basic Medical Sciences, Tianjin Medical University, 300070 Tianjin, China. [5]Genome Institute of Singapore (GIS), Agency for Science, Technology and Research (A*STAR), 60 Biopolis St, Genome, Singapore 138672, Republic of Singapore. [6]Department of Clinical Laboratory, Tianjin Medical University General Hospital, 300052 Tianjin, China. ✉E-mail: haifangyin@tmu.edu.cn

against CP05-immobilized murine myotube-derived exosomes (Gao et al, 2018), and identified an orthogonal exosome-binding DNA aptamer (named Exosomal Anchor Aptamer—EAA). EAA showed strong binding affinity to exosomes from different origins and enabled efficient loading of coagulation factor IXa (FIXa) antagonist RNA aptamer (9.3t), thrombin DNA aptamer inhibitor NU172 (Keefe et al, 2010; Rusconi et al, 2002) or Duchenne Muscular Dystrophy (DMD) phosphorodiamidate morpholino oligomer (PMO) drug, an antisense oligonucleotide (AO) targeting at murine *dmd* exon 23 (Bauman et al, 2009), on murine myotube-derived exosomes via annealing or direct synthesis. Systemic administration of $EXO_{EAA-PMO}$ (PMO annealed to EAA via a complementary sequence) at the PMO dose of 25 mg/kg per week elicited therapeutic levels of dystrophin, resulting in functional improvements in *mdx* mice. Taken together, our study demonstrates that EAA enables efficient loading of different nucleic acid drugs on exosomes without complicated conjugation or modification, thus providing a facile and generalizable strategy for loading nucleic acid therapeutics on exosomes and accelerating clinical development of exosomes as delivery vehicles for nucleic acid drugs, potentially lowering therapeutic costs for patients.

# Results

## Identification of the lead exosome anchor aptamer via SELEX

To screen DNA aptamers specifically binding to exosomes, we immobilized murine myotube-derived exosomes (Fig. EV1A,EV1B) with CP05 (Fig. EV1C), a CD63-binding exosomal anchor peptide identified from our previous study (Gao et al, 2018), and screened against a DNA aptamer library via SELEX as illustrated in Fig. 1A. The bound DNA aptamers were eluted and amplified at each round prior to next round of screening and FITC signals increased each round as detected by FACS, and peaked at round 9 and 10 (Fig. 1B), with no discernible difference between round 9 and 10 (Appendix Fig. S1). Thus we selected round 9 for subsequent high-throughput sequencing, with one candidate aptamer (HG-01) showing the highest frequency (Fig. 1C; Appendix Table S1). Alignment with Aptamer Database Apta-Index™ (https://www.aptagen.com/apta-index/) revealed no similarity to any known motif. HG-01 showed greater binding to murine myotube-derived exosomes than HG-02 or HG-09 across different concentrations, with the difference being much greater at low concentrations (Fig. 1D). To identify the core motif of HG-01, we designed a series of truncations and the variant with primer ends removed ($T_{1-18\&59-76}$) showed equivalent binding affinity to full-length HG-01 at two different concentrations, whereas further shortening compromised binding affinity (Fig. 1E,F), indicating that the 42mer sequence is the core-binding motif for exosomes, and thus this variant was named as Exosomal Anchor Aptamer (EAA).

## EAA outperforms CD63 DNA aptamer in binding to exosomes

As predicted with UNAFold (http://www.unafold.org/), a software for nucleic acid folding and hybridization (Markham and Zuker, 2008), EAA exhibited a typical stem-loop secondary structure, similar to CD63 DNA aptamer (Wu et al, 2021), though with a

higher GC content in stem (Fig. 2A). Measurement of binding affinity of EAA to murine myotube-derived exosomes with FACS showed an equilibrium dissociation constant ($K_D$) value of $51.38 \pm 5.12$ nM, which shows a stronger binding affinity than CD63 DNA aptamer (Wu et al, 2021) ($187.45 \pm 8.32$ nM) (Fig. 2B). Likewise, higher binding affinity was observed for EAA with exosomes from mouse serum (Fig. EV2A). To accurately measure the number of aptamers per exosome, we used single-molecule super-resolution fluorescent imaging system to analyze the co-localization of aptamers with exosomes (Chen et al, 2016). Strikingly, significantly higher numbers of Cy5-labeled EAAs were bound to single exosome derived from murine myotubes, with up to $40 \pm 0.8$ per exosome, compared to CD63 DNA aptamer ($12 \pm 0.7$) when $2.5 \times 10^9$ particle exosomes were incubated with 1 μM EAA or CD63 DNA aptamer (Fig. 2C). Notably, EAA binding did not alter the morphology and size of murine myotube-derived exosomes as demonstrated by transmission electron microscopy (TEM) and nanoparticle tracking analysis (NTA) (Fig. EV2B,EV2C). Consistently, a significantly greater binding efficiency was observed for EAA with exosomes from human or murine cells or rat and mouse serum across different concentrations, than CD63 DNA aptamer (Figs. 2D,E and EV2D), confirming the stronger binding affinity of EAA to exosomes than CD63 DNA aptamer irrespective of sources. Altogether, these findings support the conclusion that EAA shows strong binding affinity to exosomes at a single molecule or population level.

## EAA-mediated NU172 loading on exosomes improves its stability and activity

To investigate whether EAA can facilitate loading of other nucleic acid aptamers on exosomes, we chose an unmodified thrombin DNA aptamer NU172 inhibitor and synthesized it with EAA, in which NU172 is currently in phase II clinical trials and suffers from rapid degradation (Keefe et al, 2010). NU172 was efficiently loaded on murine myotube-derived exosomes via EAA ($EXO_{EAA-N}$) at two different concentrations (Fig. 3A,B). Measurement of intact EAA-NU172 in serum at different timepoints revealed that EAA-NU172 remained intact in the form of $EXO_{EAA-NU172}$ at 4 h after incubation, whereas no trace of EAA-NU172 was found in the absence of exosomes after 2 h of incubation (Fig. 3C), indicating the protective effect of exosomes on NU172 from rapid degradation in serum in vitro. Concordantly, significantly extended clotting time was observed with $EXO_{EAA-N}$ compared to EAA- NU172 alone when thrombin was added to plasma in vitro, and no effect was found for exosomes alone (Fig. 3D). Notably, comparable activities were exhibited for NU172 and EAA-NU172 in vitro (Fig. EV3A), indicating that EAA does not interfere with the activity of NU172. Significantly reduced clotting and improved blood flow to a level comparable to sham controls was observed with $EXO_{EAA-N}$ in contrast to clotting seen with EAA-NU172 alone (Fig. 3E,F) when $EXO_{EAA-N}$ were intravenously administered to mice with arterial injury induced by ferric chloride ($FeCl_3$), a widely used model for occlusive thrombosis (Li et al, 2013). We also tested the feasibility of loading anti-coagulation factor IXa (FIXa) RNA aptamers (9.3t) (Rusconi et al, 2002) by adding an RNA linker to the 5' end of FIXa RNA aptamer that could anneal to a DNA linker sequence attached to the 3' end of EAA (Fig. 3G). As expected, FIXa RNA aptamers were efficiently loaded on murine myotube-derived exosomes via

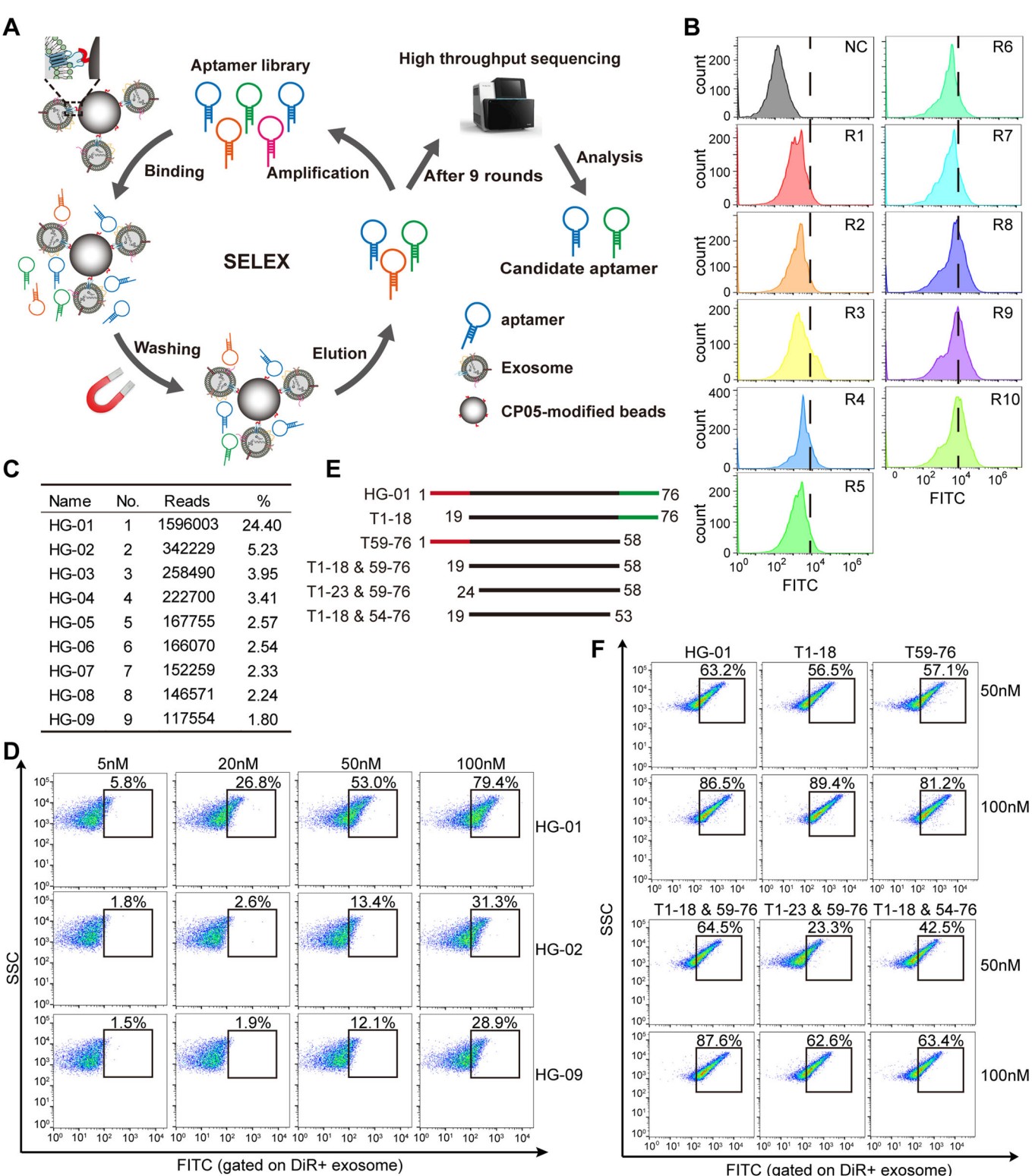

**Figure 1. Screening and identification of Exosome Anchor Aptamer (EAA).**

(A) Diagram for the SELEX experimental design. (B) Flow cytometric analysis to assess the enrichment of DNA aptamers for each round of screening. (C) Table of the top nine candidate aptamers in the order of reads by high-throughput sequencing. (D) Flow cytometric analysis of binding efficiency of candidate aptamers to murine myotube-derived exosomes at different concentrations ($n = 3$). Exosomes were labeled with DiR and candidate DNA aptamers were labeled with FITC. (E) Schematic for the design of different truncations of HG-01. (F) Flow cytometric analysis of binding efficiency of different variants to murine myotube-derived exosomes across different concentrations ($n = 3$). Data represent different numbers ($n$) of biological replicates. Source data are available online for this figure.

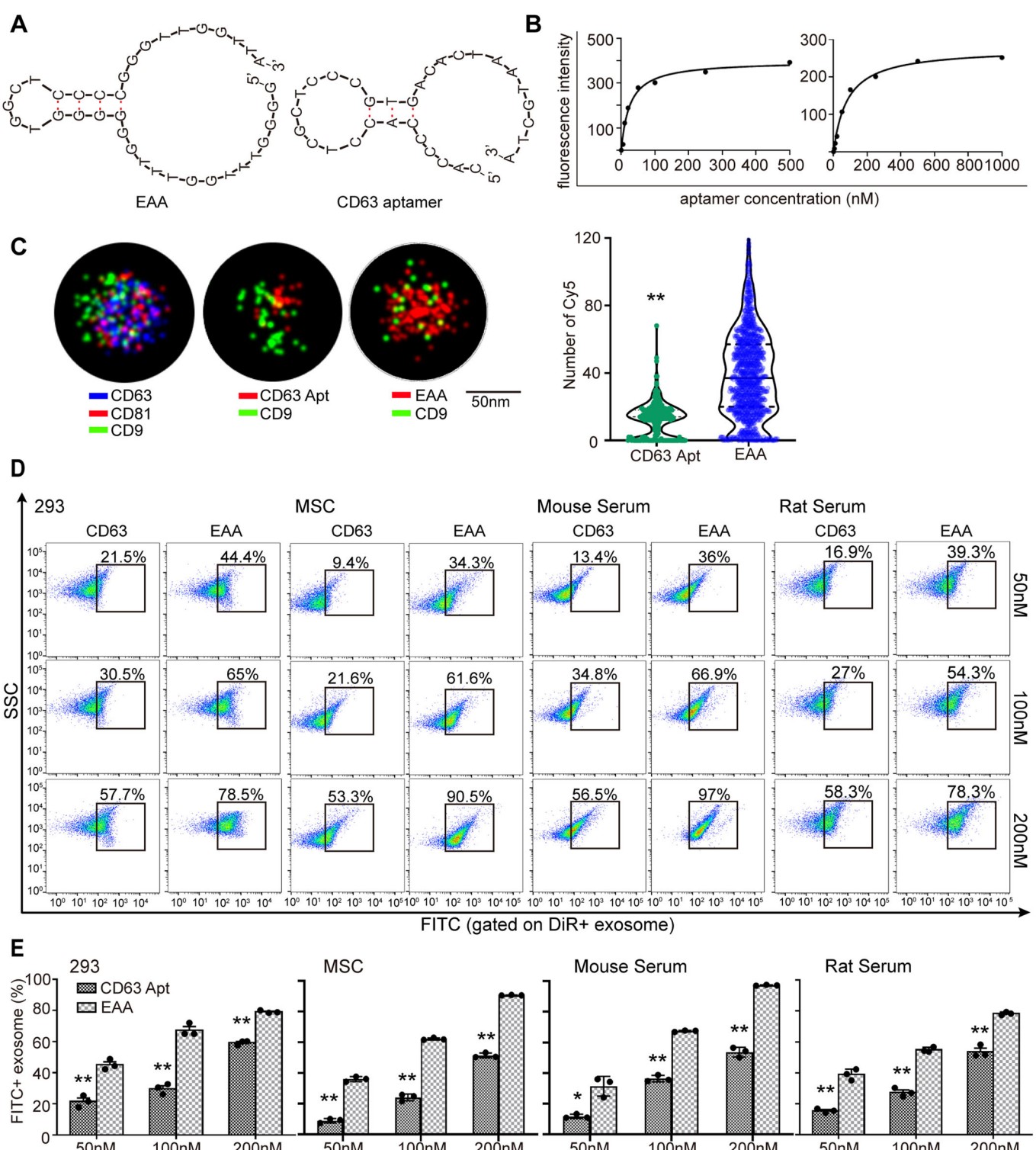

EAA after annealing at 1:1 molar ratio across different concentrations (Figs. EV3B and 3H). Collectively, these data demonstrate that EAA enables facile and efficient loading of DNA and RNA aptamers on exosomes, and loading of thrombin DNA aptamer inhibitors on exosomes results in prolonged stability and activity in vitro and in an occlusive thrombosis model in vivo.

## EAA enables PMO loading on exosomes with enhanced activity in *mdx* mice

We next wished to explore whether EAA is able to load nucleic acid analogs to exosomes. Toward this end, we took advantage of the intrinsically stable property of DMD exon-skipping PMO drugs,

◄ **Figure 2. Evaluation of the binding affinity of EAA to exosomes from different sources.**

(A) Predicted secondary structure of EAA and CD63 aptamer with UNAFold. (B) Measurement of binding affinity of EAA and CD63 aptamers to murine myotube-derived exosomes with flow cytometry. (C) Representative images and quantitative analysis for single exosomes bound with EAA ($n = 1193$) or CD63 ($n = 221$) aptamers via super-resolution microscopy (scale bar = 50 nm). Murine myotube-derived exosomes were stained with CD63, CD81, or CD9 antibodies and EAA or CD63 aptamers were labeled with FITC. (D, E) Flow cytometric (D) and quantitative analysis (E) of binding efficiency of EAA or CD63 aptamers to exosomes across different concentrations ($n = 3$). Exosomes from human 293 cells and mesenchymal stem cells (MSC), and mouse and rat serum were used. Data information: Data represent different numbers ($n$) of biological replicates. (C) Data are presented as violin plot with lines at median and quartiles; statistical significance was determined using Mann–Whitney rank-sum test. (E) The data with error bars are shown as mean ± SEM; statistical significance was determined using two-tailed $t$ test. *$P < 0.05$, **$P < 0.001$. Source data are available online for this figure.

which have been clinically approved for the treatment of DMD with limited systemic efficacy (Hammond et al, 2021), and loaded DMD PMO drugs to murine myotube-derived exosomes by incorporating a DNA sequence complementary to PMO (PMOC) into EAA (Fig. 4A). Incorporation of PMOC did not affect binding of EAA to murine myotube-derived exosomes (Fig. EV4A) and PMO was effectively annealed to EAA-PMOC across different molar ratios (Fig. EV4B). Importantly, PMO was efficiently loaded on myotube-derived exosomes to form $EXO_{EAA-PMO}$ (Fig. 4B) and $EXO_{EAA-PMO}$ were readily taken up by murine myoblasts compared to PMO alone (Fig. 4C), demonstrating the ability of exosomes to promote cell internalization of PMOs. Consistently, $EXO_{EAA-PMO}$ induced significantly greater numbers of dystrophin-positive fibers and levels of exon-skipping and dystrophin restoration than PMO alone in *mdx* mice after intramuscular injection (Fig. 4D–F), confirming that EAA-mediated PMO loading on exosomes enhances its cellular internalization and activities in muscle cells in vitro and in vivo.

## $EXO_{EAA-PMO}$ induce effective dystrophin expression and functional improvement in *mdx* mice

To evaluate the systemic efficacy of $EXO_{EAA-PMO}$, we intravenously administered $EXO_{EAA-PMO}$ at the PMO dose of 25 mg/kg/week for 3 weeks in *mdx* mice. More uniform and widespread dystrophin-positive fibers and significantly higher levels of dystrophin protein were found in peripheral muscles, except for the heart, of *mdx* mice treated with $EXO_{EAA-PMO}$ than PMO alone (Fig. 5A,B). Correlating with therapeutic levels of dystrophin restoration in $EXO_{EAA-PMO}$-treated *mdx* mice, functional and pathological improvements were demonstrated by significantly improved muscle strength (Fig. 5C) and significant decline in serum creatine kinase (CK) levels (Fig. 5D), usually elevated in *mdx* mice due to leakage from diseased muscle tissues (Ozawa et al, 1999), compared to PMO alone and age-matched untreated *mdx* controls. Remarkably, measurement of PMO in different muscle tissues revealed that significantly increased levels of PMO in gastrocnemius and quadriceps of *mdx* mice treated with $EXO_{EAA-PMO}$ compared to PMO (Fig. 5E), indicating that exosomes promote delivery of PMO to muscle. Analysis of serum biochemical indices including aspartate aminotransferase (AST), alanine aminotransferase (ALT), urea and blood urea nitrogen (BUN) revealed that levels of AST and ALT elevated in *mdx* mice (Brazeau et al, 1992) were significantly reduced in *mdx* mice treated with $EXO_{EAA-PMO}$ compared to PMO and untreated *mdx* controls (Fig. 5F), whereas no change was found on levels of BUN and urea (Fig. EV5A). Consistently, histological examination on liver and kidney exhibited no abnormal morphological changes (Fig. EV5B). Notably, inflammation and fibrosis were alleviated in muscles

from *mdx* mice treated with $EXO_{EAA-PMO}$ as demonstrated by H&E and mass' trichrome staining (Figs. 5G and EV5C). These findings demonstrate that exosomes augment PMO activities by enhancing delivery to muscle, resulting in the induction of therapeutic levels of dystrophin and functional and pathological improvements without any delectable toxicity in *mdx* mice.

## Discussion

Oligonucleotide drugs have reached fever pitch after decades of silence, with numerous drugs being approved or in clinical trials (Kulkarni et al, 2021). Safe and effective delivery of nucleic acid therapeutics in vivo has thus become crucial technology gaps to clinical deployment (Hammond et al, 2021). Exosomes have been extensively tested in clinical trials for a myriad of diseases (Tan et al, 2024), and also being actively exploited for treating DMD either as therapeutics or delivery vehicles (Yedigaryan and Sampaolesi, 2023). Although most studies demonstrated therapeutic effects on skeletal muscles with unmodified or modified exosomes from different sources (Gao et al, 2018; Leng et al, 2021; Ran et al, 2020; Sandona et al, 2020), Marban and colleagues reported that exosomes from clinical stage cardiosphere-derived cells improved both cardiac and skeletal muscle functions in DMD models (Aminzadeh et al, 2018; Rogers et al, 2019), showing the potential and clinical applicability of exosomes for treating DMD. Importantly, it seems that the source of exosomes might have impact on tissue distribution, with exosomes from cardiosphere-derived cells showing superior heart-targeting property to exosomes from other sources. We speculated that cardiogenic moieties on exosomes from cardiosphere-derived cells might be primarily responsible for targeting the heart though other possibilities cannot be excluded. Functional and tissue-targeted modification of exosomes from clinical stage cells such as cardiosphere-derived cells (in Phase III clinical trial NCT05126758) or other stem cells would further increase efficacy in target tissues. Worth mentioning, PMOs are expensive to manufacture and even more expensive to conjugate. If there were methods to increase efficacy per PMO molecule injected, it could make a big difference in the affordability of the approach. Exosomes can extend circulatory half-life of PMO tremendously, lowering required dosage, but if loading onto exosomes and targeting exosomes were prohibitively expensive, that would defeat any savings arising from reduced dosage. Previously, we had identified CP05 which was a peptide that can be used to modify exosome surface (Gao et al, 2018), but conjugating peptides to nucleic acids is still more expensive than lengthening the synthesis length of an oligonucleotide, thus EAA is a new tool in the arsenal to help enhance exosomes to a point where

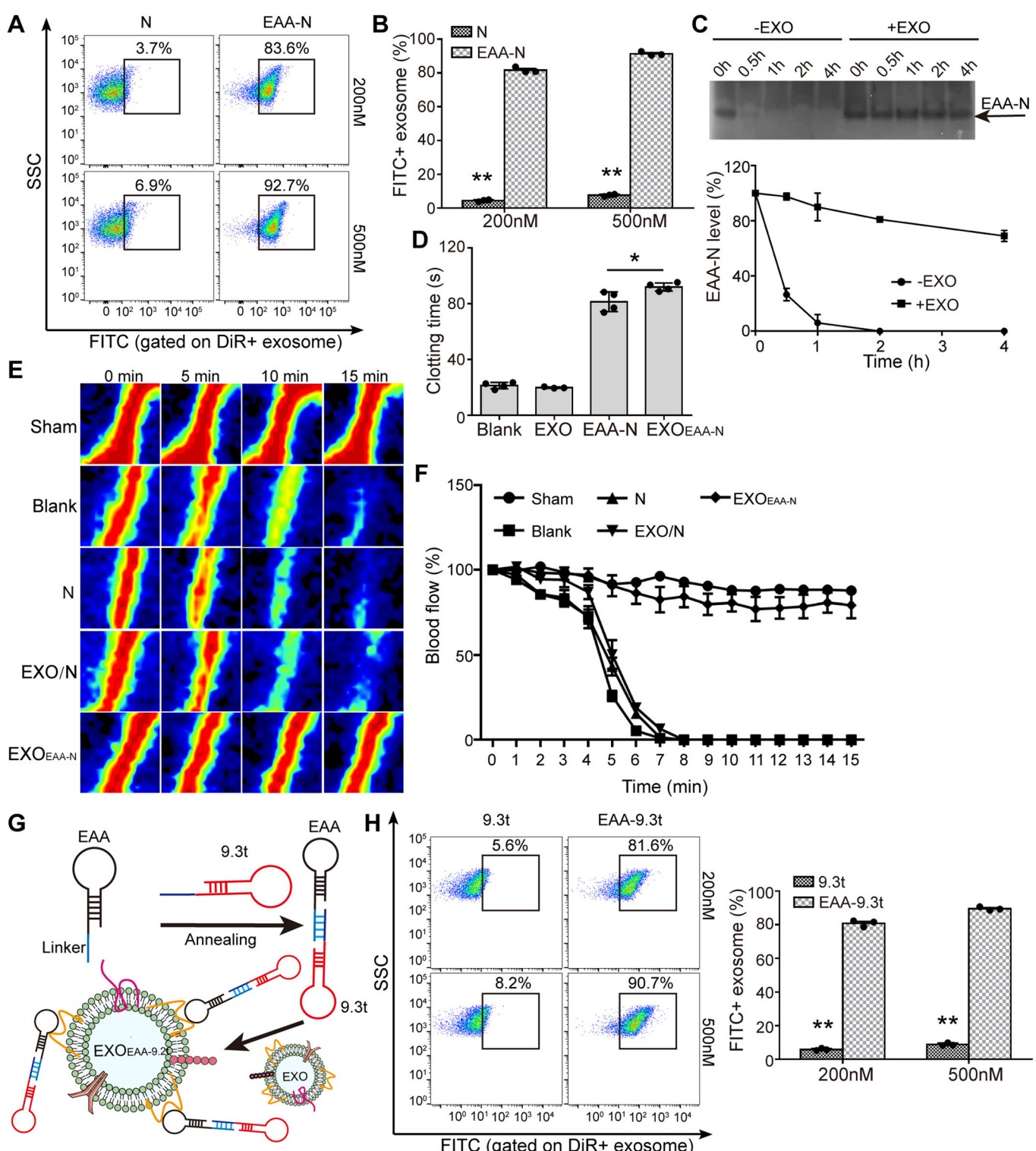

PMOs could be dosed at very low doses, enabling greater access to therapy for DMD patients, however often they need to take the drugs. Also exosomes were shown to be able to deliver Cas9 ribonucleoprotein complexes to liver (Wan et al, 2022) and repeated administration might not be necessary for DMD patients in future if Cas9 and single guide (sg)RNA can be co-delivered to

muscle and heart efficiently. Thus, new approaches for efficient loading of nucleic acids to exosomes will accelerate their clinical use. Nucleic acid aptamers have been extensively employed in diagnostics as probes for capture and detection of exosomes and as therapeutics (Kumar Kulabhusan et al, 2020; Wu et al, 2021; Zhu et al, 2020), but have not been utilized as anchors for loading

**Figure 3. Systemic evaluation of DNA or RNA aptamers loaded on exosomes via EAA in vitro and in vivo.**

(A, B) Flow cytometric (A) and quantitative (B) analysis to assess the binding efficiency of EAA-NU172 to murine myotube-derived exosomes across different concentrations ($n = 3$). Exosomes were labeled with DiR and EAA- NU172 (EAA-N) or NU172 (N) aptamers were labeled with FITC. (C) Representative silver staining and quantitative analysis of EAA-NU172 loaded on exosomes at different timepoints ($n = 3$). Murine myotube-derived exosomes were used. (D) Measurement of plasma clotting time in the presence of EXO ($n = 3$), EAA-N or EXO$_{EAA-N}$ ($n = 4$) and untreated controls ($n = 4$). Fresh anti-coagulated rat plasma and murine myotube-derived exosomes were used for the assay. (E) Representative real-time imaging of blood flow in arterial thrombosis mice treated with NU172 (N), the mixture of NU172 and exosomes (EXO/N) or EXO$_{EAA-N}$ ($n = 4$). (F) Quantitative analysis of arterial blood flow of treated thrombosis mice with transonic T400 at different timepoints. Mice were treated with NU172 (N) ($n = 5$), the mixture of NU172 and exosomes (EXO/N) ($n = 5$), EXO$_{EAA-N}$ ($n = 10$), untreated ($n = 5$) or sham controls ($n = 6$). Blank refers to untreated controls. (G) Schematic for the design of loading 9.3t RNA aptamers on murine myotube-derived exosomes via EAA. (H) Flow cytometric and quantitative analysis to assess the binding efficiency of EAA-9.3t RNA aptamers to murine myotube-derived exosomes ($n = 3$). Data information: Data represent different numbers ($n$) of biological replicates. The data with error bars are shown as mean ± SEM. (B, H) Statistical significance was determined using a two-tailed $t$ test. (D) Statistical significance was determined using one-way ANOVA on ranks test; *$P < 0.05$, **$P < 0.001$. Source data are available online for this figure.

nucleic acid drugs to exosomes. Here, we demonstrated for the first time that exosomal anchor aptamer EAA enables facile and efficient loading of nucleic acid drugs of different chemical backbones including DNA and RNA aptamers, and PMOs on exosomes, resulting in improved serum stability, cellular internalization, target tissue accumulation and activity in vitro and in vivo.

In this study, we immobilized murine myotube-derived exosomes via our proprietary exosomal anchor peptide CP05 (Gao et al, 2018) and used the entire exosome as target rather than any specific exosome protein to increase the success rate of screening. CP05 has been extensively used for loading of different moieties including peptides, immunoadjuvants, and nucleic acids on exosomes (Dong et al, 2021; Gao et al, 2018; Ma et al, 2022; Wang et al, 2021; Zou et al, 2021; Zuo et al, 2022). Although CP05 is effective at loading peptides and can mediate efficient loading of neutral PMO drugs on exosomes, the conjugation of peptide to other negatively charged nucleic acid drugs such as aptamers, siRNAs and other non-neutral AOs is quite challenging and costly. In clear contrast, EAA, as a DNA aptamer, can load nucleic acid drugs irrespective of chemistry on exosomes in a more facile and cost-effective manner via direct synthesis or annealing as demonstrated in our current study. CP05 and EAA can complement each other in targeted nucleic acid drug delivery by combining the superiority of CP05 to load tissue-targeting peptides and EAA for nucleic acid drugs on exosomes.

Although the unmodified EAA was selected against murine myotube-derived exosomes, EAA shows strong binding affinity to exosomes of different sources. Further modification and optimization will probably improve the efficacy and warrants future studies. Compared to the CD63-binding aptamer, exosome-binding affinity and copies per exosome is significantly higher for EAA. We speculated that it is likely due to the abundance of its targets on exosomes as EAA was identified against intact exosomes rather than a single protein as CD63 DNA aptamer (Zhou et al, 2016). Digestion of proteins on the surface of exosomes with trypsin significantly compromised the binding of EAA but not by blocking common protein biomarkers on the surface of exosomes including CD63, CD9, or CD81, individually or in combination with antibodies. These findings suggest that EAA specifically binds to protein on the surface of exosomes but not to CD63, CD9 or CD81. However, the identification of binding partners for EAA is challenging due to the highly complex surface molecules of exosomes. More extensive studies are warranted to identify the receptor(s) of EAA in future.

In summary, we demonstrated that EAA enables facile and efficient loading of various nucleic acid drugs on exosomes, resulting in enhanced stability, cellular internalization, target tissue

accumulation and activities in vitro and in vivo, thus providing a generalizable strategy for loading nucleic acid drugs on exosomes, potentially enabling a route towards more effective and affordable exosome-based oligonucleotide therapies.

# Methods

## Animals and injections

Adult *mdx* (6–8-week old) (purchased from The Jackson laboratory, USA) and age-matched *C57BL/6* mice (purchased from Beijing Vital River Laboratory) were used in all experiments (numbers of mice per group specified in the animal experiment and corresponding figure legends, no gender preference). Adult rats for the plasma clotting experiment were purchased from Si Pei Fu (Beijing, China). The animal experiments were carried out in the Animal unit, Tianjin Medical University (Tianjin, China), according to procedures authorized by the institutional ethical committee (Permit Number: SYXK 2023-0004). For the establishment of arterial thrombosis mouse model, thrombosis in the carotid artery was induced by applying a 5% ferric chloride solution in mouse carotid artery as previously reported (Rusconi et al, 2004). Briefly, NU172, the mixture of NU172 and murine myotube-derived exosomes (EXO/N, 1:1) or NU172 loaded on exosomes (EXO$_{EAA-N}$) (1:1 mass ratio) at the NU172 or EAA-NU172 dose of 0.1 nmol/g (body weight) were injected into arterial thrombosis mice via tail vein. Vascular blood flow was monitored with small animal laser Doppler (PeriCam PSI, Switzerland) and blood flow measurement devices (Transonic T400, USA) for 15 min. For the real-time imaging of blood flow with small animal laser Doppler, four mice per group were used. For quantitative analysis of arterial blood flow with blood flow measurement devices, different numbers of mice for each group were used with NU172 (N) ($n = 5$), the mixture of NU172 and exosomes (EXO/N) ($n = 5$), EXO$_{EAA-N}$ ($n = 10$), untreated ($n = 5$) or sham controls ($n = 6$). EXO$_{EAA-PMO}$ were prepared by mixing PMO and EAA at the molar ratio of 2:1, followed by the addition of murine myotube-derived exosomes at the mass ratio of 2:1 and incubated for 2 h at 4 °C. For local intramuscular study, 1 μg (0.12 nmol) PMO or EXO$_{EAA-PMO}$ was dissolved in Dulbecco's phosphate-buffered saline (DPBS) and injected into tibialis anterior (TA) muscle of *mdx* mice (5 mice per group). For the systemic study, equimolar PMO (5 mice) or EXO$_{EAA-PMO}$ (4 mice) in DPBS was repeatedly injected at the PMO dose of 25 mg/kg/week (2.97 μmol/kg/week) for 3 weeks in *mdx* mice intravenously. Mice were sacrificed by CO$_2$ inhalation at 2 weeks after the last injection, and muscles and other tissues were snap-frozen in liquid nitrogen-cooled isopentane and stored at −80 °C.

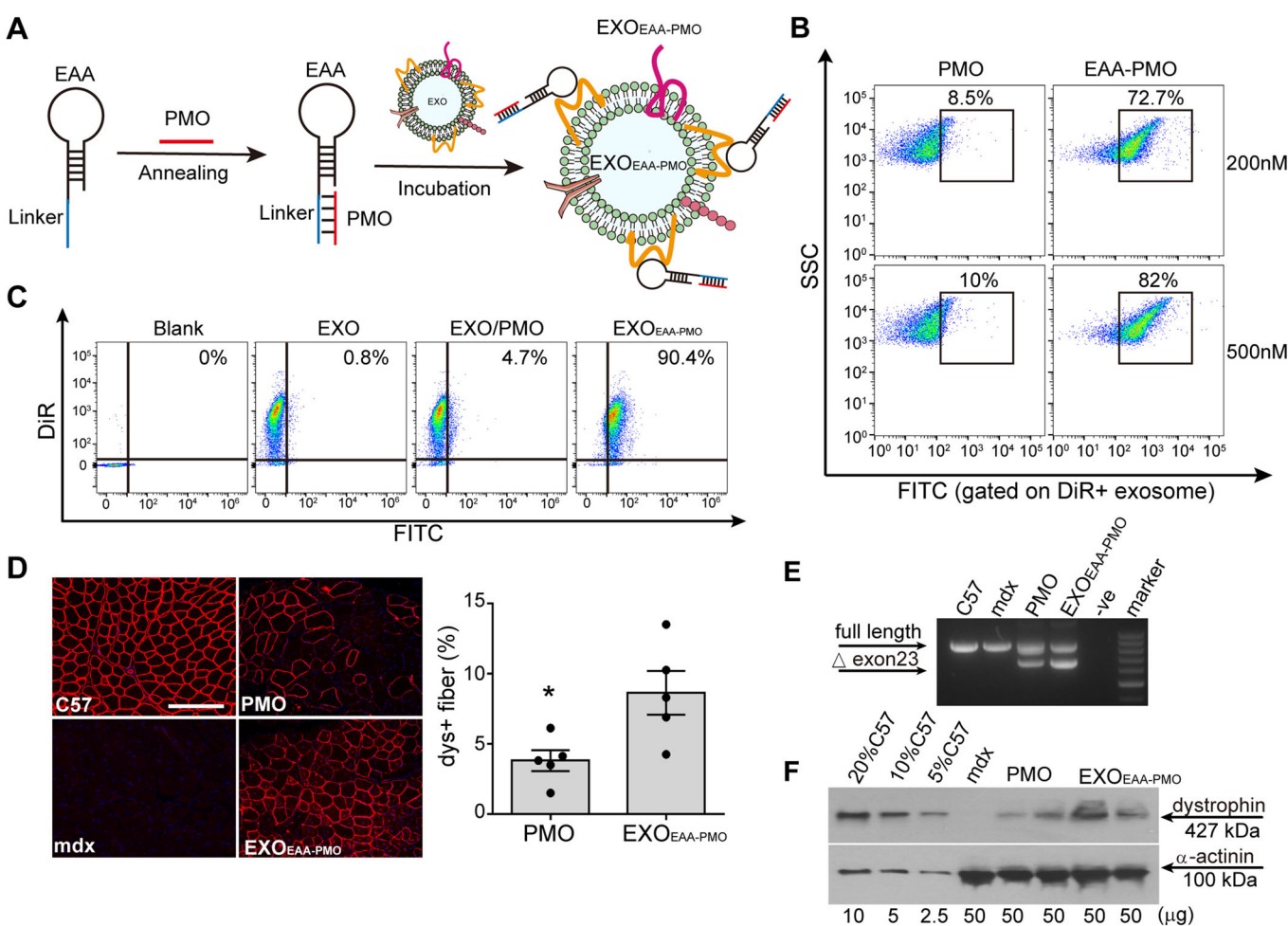

**Figure 4. Assessment of cellular uptake and activities of PMO loaded on exosomes via EAA in vitro and in *mdx* mice intramuscularly.**

(A) Diagram for the loading of PMO on murine myotube-derived exosomes via EAA (EXO$_{EAA-PMO}$). (B) Flow cytometric analysis to assess the binding efficiency of EAA-PMO or PMO to exosomes ($n = 3$). (C) Measurement of cellular uptake of EXO$_{EAA-PMO}$, the mixture of PMO and exosomes (EXO/PMO) or exosomes alone in murine myoblasts 24 h after incubation. (D) Immunostaining and quantitative analysis of dystrophin-positive (dys⁺) myofibres in tibialis anterior (TA) muscles of *mdx* mice treated with PMO or EXO$_{EAA-PMO}$ (scale bar = 100 μm) ($n = 5$). (E) RT-PCR to determine the level of exon-skipping in TA muscles from treated *mdx* mice. Δexon 23 for exon 23 skipped bands. -ve means negative control. (F) Representative Western blot image to show dystrophin restoration in TA muscles of treated *mdx* mice. 2.5, 5, and 10 μg total protein from *C57BL/6* and 50 μg from untreated and treated *mdx* muscle samples were loaded, respectively. α-actinin was used as the loading control. Data information: Data represent different numbers ($n$) of biological replicates. The data with error bars are shown as mean ± SEM. Statistical significance was determined using two-tailed *t* test. *$P < 0.05$. Source data are available online for this figure.

## Cell culture

Human embryonic kidney 293T, murine fibroblast NIH 3T3, murine C2C12 cells (myoblasts) were kept in house and cultured as previously reported (Ferri et al, 2009; Lin et al, 2014; Rahimi et al, 2022). C2C12 cells were grown at 37 °C in 5% $CO_2$ in Dulbecco's modified Eagle's medium (DMEM) supplemented with 10% fetal bovine serum (FBS) and 1% penicillin and streptomycin. Myotubes were obtained from confluent C2C12 seeded in gelatin-coated 12-well plates after 3 days of serum deprivation at 37 °C under a 5% $CO_2$ atmosphere in DMEM with 2% horse serum (Hyclone, USA). Cell supernatant of human mesenchymal cells was kindly provided by SHUNHO CELL BIOLOGY TECHNOLOGY (TIANJIN) Corporation (Tianjin, China).

## Preparation and purification of exosomes

Cell culture medium was sequentially centrifuged at $1000 \times g$ for 10 min, followed by $10,000 \times g$ for 30 min. Serum was obtained from *C57BL/6* wild-type mice or Sprague–Dawley rats and centrifuged at $3000 \times g$, $6000 \times g$, and $10,000 \times g$ for 30 min sequentially to remove cell debris. The supernatant was collected and filtered with a 0.22 μm filter (Millex, USA), followed by ultracentrifugation at $100,000 \times g$ for 1 h to pellet exosomes and further purified by sucrose density gradient centrifugation. Briefly, exosomes were layered on a linear sucrose gradient (0.65, 0.85, 1.05, 1.25, 1.45, and 1.65 M sucrose (Sigma, USA). The gradients were centrifuged for 16 h at $100,000 \times g$ at 4 °C. Six fractions from 0.65 to 1.65 M sucrose gradients were collected and ultracentrifuged at $100,000 \times g$ for 1 h at 4 °C to pellet exosomes. Exosome pellets were

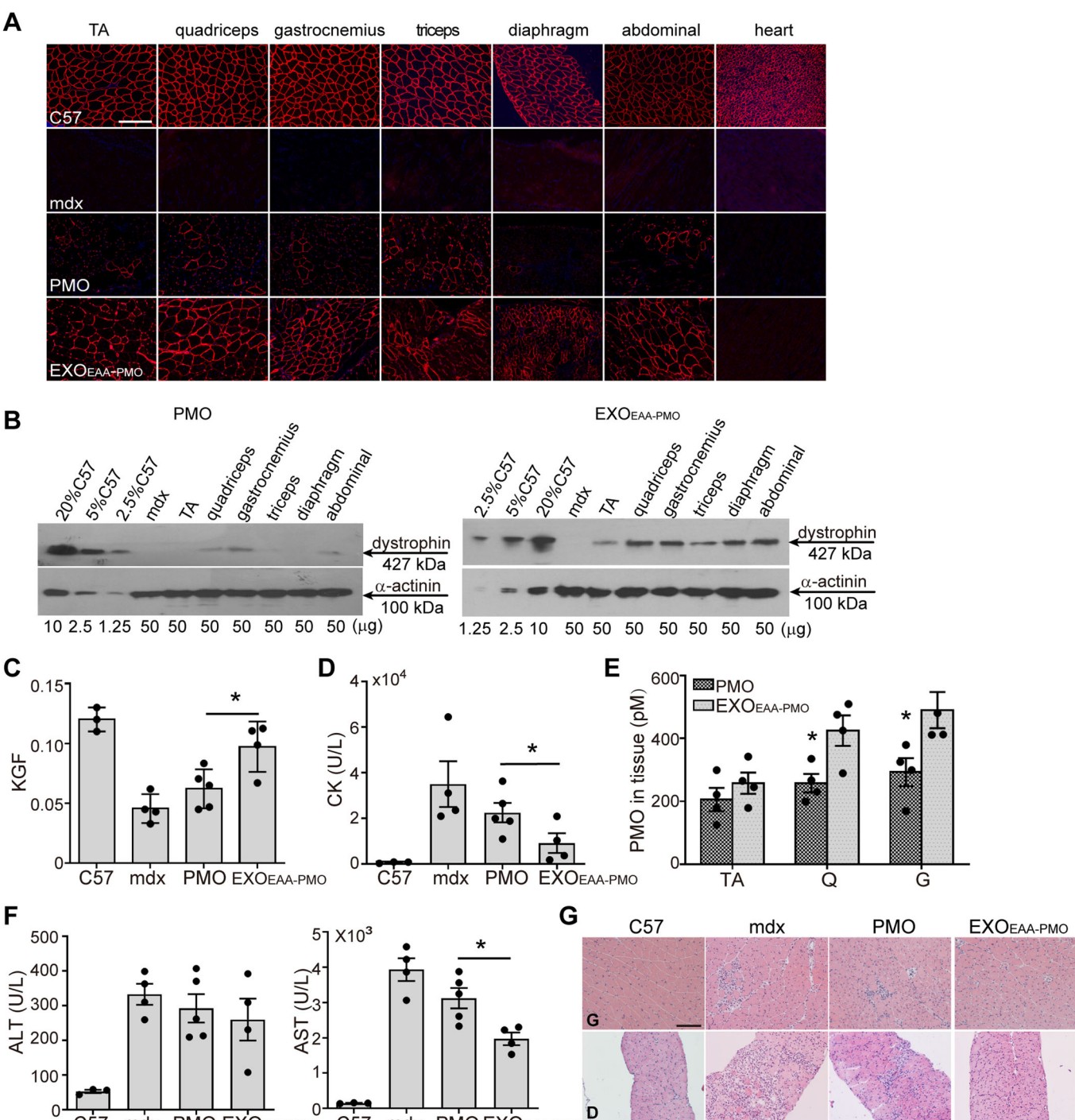

washed in a large volume of PBS and recovered by centrifugation at 100,000 × g for 1 h. The total protein concentration of exosomes was quantified by the Bradford assay (Sangon Biotech, China).

## Characterization of exosomes

The morphology of serum exosomes was visualized with a high-resolution transmission electron microscope (TEM, Hitachi HT7700, Tokyo, Japan) as described previously (Gao et al, 2018). Briefly, the resuspended exosomes were diluted into PBS (1 μg/μl) and mixed with

an equal volume of 4% paraformaldehyde (PFA). Exosomes were adsorbed onto a glow-discharged, carbon-coated formvar film attached to a metal specimen grid. Excess solution was tipped off, and the grid was immersed with a small drop (50 μl) of 1% glutaraldehyde for 5 min followed by washing with 100 μl distilled water for eight times (2 min each time). Subsequently, the grid was transferred to 50 μl uranyl-oxalate solution (pH7.0) for 5 min and then 50 μl methylcellulose-uranyl acetate (100 μl 4% uranyl acetate and 900 μl 2% methylcellulose) for 10 min at 4 °C. The excess solution was blotted off, and the sample was dried and examined with TEM.

**Figure 5. Systemic investigation of EXO_EAA-PMO in adult *mdx* mice.**

EXO_EAA-PMO at the PMO dose of 25 mg/kg were administered into adult *mdx* mice for three times weekly intravenously and tissues were harvested 2 weeks after last injection. (A) Immunostaining of dystrophin-positive myofibres in body-wide muscles of *mdx* mice treated with PMO or EXO_EAA-PMO (scale bar = 100 μm). (B) Representative Western blot image to show dystrophin restoration in body-wide muscles of treated *mdx* mice. 1.25, 2.5, and 10 μg total protein from *C57BL/6* and 50 μg from untreated and treated *mdx* muscle samples were loaded, respectively. α-actinin was used as the loading control. (C) Muscle function was assessed to determine the physical improvement of *mdx* mice treated with PMO (n = 5) or EXO_EAA-PMO (n = 4), untreated *mdx* (n = 4) and wild-type controls (n = 3) with the grip strength test. Two-tailed *t* test was used for the statistical analysis between PMO and EXO_EAA-PMO. (D) Measurement of serum creatine kinase (CK) levels in *mdx* mice treated with PMO (n = 5) or EXO_EAA-PMO (n = 4), untreated *mdx* controls (n = 4) and wild-type mice (n = 3). (E) Measurement of PMOs in muscle tissues from treated *mdx* mice (n = 4). TA tibialis anterior, Q quadriceps, G gastrocnemius. (F) Analysis of serum levels of liver enzymes in *mdx* mice treated with PMO (n = 5) or EXO_EAA-PMO (n = 4), untreated *mdx* controls and wild-type mice (n = 3). (G) H&E staining of peripheral muscle tissue sections from *mdx* mice treated with PMO or EXO_EAA-PMO, untreated *mdx* and *C57BL/6* controls (scale bar = 200 μm). Data information: Data represent different numbers (n) of biological replicates. The data with error bars are shown as mean ± SEM. (C–F) Statistical significance was determined using two-tailed *t* test. *P < 0.05. Source data are available online for this figure.

## Nanoparticle tracking analysis (NTA)

Nanoparticle tracking analysis (NTA) was performed with NS300 nanoparticle analyser (NanoSight, Malvern, UK) to measure the size distribution of exosomes and aptamer-painted exosomes as described previously (Gao et al, 2018). For all our recordings, we used a camera level of 13–14 and automatic functions for all post-acquisition settings except for the detection threshold, which was fixed at 5. All samples were diluted in 0.22-μm filtered PBS at the ratio of 1:100 and 1:1000 to achieve a particle count of between $1 \times 10^8$ and $1 \times 10^9$ per ml. The camera focus was adjusted to make the particles appear as sharp dots. Using the script control function, three 60-second videos for each sample were recorded, incorporating a sample advance and a 5-second delay between each recording.

## Systematic evolution of ligands by exponential enrichment (SELEX)

The single-stranded (ss) DNA aptamer library and primers (Appendix Table S1) were purchased from Sangon Biotech (Shanghai, China) and SELEX was conducted as reported previously (Sefah et al, 2010). The SELEX was screened against murine myotube-derived exosomes immobilized with CP05-coated beads. Briefly, exosomes (50 μg) were incubated with CP05-coated magnetic beads (20 μl) (kindly provided by Dr Yiqi Seow, Genome Institute of Singapore (GIS), A*STAR, Singapore) for 30 min at 4 °C. The ssDNA library (10nmol) was incubated with exosome-bound magnetic beads for 1 h at 4 °C and unbound DNA was removed by washing with PBS for five times briefly. The bound DNA was eluted by heating at 95 °C for 10 min and amplified. After the first round of selection, the recovered ssDNA pool was amplified. The cycling conditions were 95 °C for 30 s, 56.3 °C for 30 s and 72 °C for 30 s for ten cycles, followed by final extension for 3 min at 72 °C. Then the PCR product was further amplified for another 6–12 cycles for the next round of selection based on the amplification rate as described previously (Sefah et al, 2010). Following amplification, the amplified double-stranded PCR products were recovered with streptavidin-coated magnetic beads (Thermo Fisher Scientific, US) and denatured with 200 mM NaOH. Subsequently, the biotin-labeled ssDNA bound to magnetic beads was removed via magnetic separation, and the FITC-labeled ssDNA in supernatant was purified with desalt column for the next round of selection. The enriched FITC-labeled ssDNA pool was analyzed with high-throughput sequencing (GENEWIZ, China) after nine rounds of selection.

## Oligonucleotides and peptides

Aptamers including HG-01, HG-02, HG-09, EAA, CD63 aptamer, EAA-NU172, EAA-linker, linker-FIXa (9.3t), EAA-PMOC, and corresponding FITC- or biotin-labeled aptamers were synthesized and purified with HPLC by GENEWIZ (Suzhou, China) (Appendix Table S1). PMOs and FITC-labeled PMOs were synthesized by GeneTools. LLC (Corvallis, Oregon, USA). PMO sequence was targeted to the murine *dystrophin* exon 23/ intron23 boundary site as reported (Alter et al, 2006) (Appendix Table S1). CP05 (CRHSQMTVTSRL) (Gao et al, 2018) peptide was synthesized by and purchased from China peptide Ltd. (Suzhou, China) with >95% purity.

## Exosome labeling

For the labeling of exosomes, purified exosomes were incubated with 1 μM fluorescent lipophilic tracer DiR (1,1-dioctadecyl-3,3,3,3-tetramethylindotricarbo-cyanine iodide) (D12731, Invitrogen, USA) at 37 °C for 30 min, followed by ultracentrifugation at 100,000×g for 60 min to remove free DiR dye as described previously (Huang et al, 2022; Wiklander et al, 2015).

## Flow cytometry

To assess the binding affinity of candidate aptamers with exosomes, FITC-labeled aptamers at different concentrations were incubated with DiR-labeled exosomes (20 μg) at 4 °C for 1 h. Unbound aptamers were removed by 100 kDa ultrafiltration filter (Millipore, USA), followed by flow cytometry analysis as described previously (Gao et al, 2018) (Appendix Fig. S2). For quantitative analysis, FITC and DiR double-positive exosomes were gated from DiR-positive exosomes. For measuring the dissociation constant ($K_D$), murine myotube-derived exosomes or exosomes derived from mouse serum (20 μg) were incubated with Cy5-labeled EAA or CD63 aptamers at 10, 20, 40, 50, 100, 500, and 1000 nM for 1 h at 4 °C, and free DNA aptamers were removed with 100 kDa ultrafiltration filter as adopted from a previous study (Gao et al, 2018). The fluorescence signals on exosomes were detected using flow cytometry and the graph was plotted with SigmaPlot (Systat Software Inc. Chicago, IL, USA) with a fitting equation as $Y = BmaxX/(Kd + X)$ (Sefah et al, 2010).

## Super-resolution microscopy for exosomes

To examine the number of aptamers binding to single exosomes, super-resolution microscopy was used and murine myotube-derived

exosomes were stained as per the manufacturer's instructions (ONI, Oxford, UK). Briefly, 2.5 μl exosomes ($1 \times 10^9$ particles/μl) were blocked with 3.5 μl blocking solution for 5 min, followed by staining with 1 μl of anti-CD9-488, anti-CD63-568 and anti-CD81-647 (provided in the kit) or 1 μM Cy5-labeled EAA or CD63 aptamers overnight at 4 °C. The stained exosomes were added to the pre-treated lanes on the plate, followed by washing with the washing buffer to remove free antibodies or aptamers. The lanes were imaged with Nanoimager (ONI, Oxford, UK), and the image processing and data analysis were performed using the EV Profiling program on an online platform (https://alto.codi.bio/).

### Serum stability assay

EAA-NU172 or $EXO_{EAA-NU172}$ ($EXO_{EAA-N}$) at the concentration of 1 μM of EAA-NU172 was incubated in PBS containing 20% fresh mouse serum at 37 °C for 0.5, 1, 2, or 4 h, followed by heating at 95 °C for 10 min. Ten percent native polyacrylamide gel was used for electrophoresis, and followed by silver staining for visualization of remaining aptamers.

### Clotting time analysis

The clotting time analysis was adopted from a previous study (Pasternak et al, 2011). Briefly, the anti-coagulated fresh rat plasma was added to a testing cup and placed in a preheated instrument at 37 °C. In parallel, a mixture of 100 μl DPBS, 1U thrombin, and 200 nM EAA-N or $EXO_{EAA-N}$ was prepared and incubated at 37 °C for 10 min. The resulting mixture was then added to the testing cup containing fresh rat plasma, and thoroughly mixed, and the coagulation time was promptly recorded using a coagulation analyzer (PUN- 2048 A, China).

### Muscle cell uptake

Murine C2C12 cells ($8 \times 10^5$) were seeded in 24-well plates overnight. FITC-labeled PMOs (5 μM) alone or annealed to EAAC (5 μM) were incubated with DiR-labeled exosomes (40 ng/μl) for 2 h at 4 °C, respectively. Exosomes (EXO), the mixture of FITC-labeled PMO with exosomes (EXO/PMO) and $EXO_{EAA-PMO}$ were added into C2C12 cells and incubated for 24 h. Subsequently, cells were washed with PBS and digested for flow cytometric analysis (BD verse, USA).

### RNA extraction and nested RT-PCR analysis

Total RNA was extracted with Trizol (Invitrogen, UK) as per the manufacturer's instructions and 400 ng of RNA template was used for 20 μl RT-PCR with OneStep RT-PCR kit (Qiagen, UK). The primer sequences for the initial RT-PCR were exon 20F0: 5'-CAGAATTCTG CCAATTGCTGAG-3' and exon 26R0: 5'-TTCTTCAGC TTTTG TGTCATCC-3' for reverse transcription from messenger RNA and amplification of complementary DNA from exons 20 to 26. The cycling conditions were 95 °C for 1 min, 55 °C for 1 min, and 72 °C for 2 min for 25 cycles. The primer sequences for the second round were exon 20F1: 5'-CCCAGTCTACCACCCTATCAGAGC-3' and exon 24R1: 5'-CCTGCCTTTAAGGCTTCCTT-3'. The cycling conditions were 95 °C for 1 min, 57 °C for 1 min and 72 °C for 1 min for 25 cycles. The products were examined by electrophoresis on a 2% agarose gel.

### Immunohistochemistry and histology

Series of 8 μm sections were examined for dystrophin with a rabbit polyclonal antibody as described previously (Gao et al, 2018). Routine hematoxylin and eosin staining was used to examine liver, kidney, and peripheral muscle morphology. Masson's trichrome staining kit (Solarbio, China) was applied for the collagen staining. Briefly, series of 8-μm sections were fixed overnight in Bouin's solution, followed by the staining with the kit as per the manufacturer's instructions.

### Protein extraction and western blot

Exosome and cell pellets were lysated in lysis buffer (125 mM Tris-HCl, pH6.8, 10% sodium dodecyl sulfate (SDS), 2 M urea, 20% glycerol and 5% β-mercaptoethanol) and subjected to 10% SDS polyacrylamide gel electrophoresis and gels were transferred to a PVDF membrane. The protein was transferred to PVDF membranes, and membranes were blocked with 5% skimmed milk and probed with primary antibodies, including mouse monoclonal antibodies Alix (1:200) and CD63 (1:200), and a rabbit polyclonal antibody Cytochrome C (1:1000, Santa Cruz Biotechnology, US). The bound primary antibody was detected by horseradish peroxidase-conjugated goat anti-mouse or anti-rabbit IgG (1:5000; Sigma, USA), respectively. To examine the expression of dystrophin, protein extraction and Western blot were carried out as previously described (Han et al, 2016). Briefly, various amounts of protein from *C57BL6* mice as a positive control and 50 μg total protein from muscles of treated or untreated *mdx* mice were used unless otherwise specified. Each experiment was performed at least three times (at least three animals).

### Functional grip strength

Treated and control mice were tested using a commercial grip strength monitor (Chatillon, UK). The procedure was performed as previously described (Han et al, 2016). Briefly, each mouse was held 2 cm from the base of the tail, allowed to grip a protruding metal triangle bar attached to the apparatus with their forepaws, and pulled gently until they released their grip. The force exerted was recorded, and five sequential tests were carried out for each mouse, averaged at 30 s apart. Subsequently, the readings for force recovery were normalized by the body weight.

### Clinical biochemistry

Serum was taken from the jugular vein immediately after sacrifice with $CO_2$ inhalation. Analysis of serum creatinine kinase (CK), aspartate aminotransferase (AST), alanine aminotransferase (ALT), serum uric acid (UA), and urea levels was performed by the clinical pathology laboratory (School of Medical Technology, Tianjin Medical University, Tianjin, China).

### ELISA for PMO

ELISA to detect the amount of PMO in muscle tissues was performed as previously described (Burki et al, 2015). Briefly, DNA probe was designed with sequences complementary to PMO (synthesized by The Beijing Genomics Institute, Beijing, China)

**The paper explained**

**Problem**

Clinical deployment of oligonucleotides requires delivery technologies that improve stability, target tissue accumulation and cellular internalization. Exosomes as nanocarriers show potential, however, an affordable and generalizable system for efficient loading and targeted delivery of oligonucleotides, irrespective of chemistry on exosomes, remain lacking.

**Results**

Our study identified an exosome-binding DNA aptamer (Exosomal Anchor Aptamer—EAA) and demonstrated that EAA shows high binding affinity to exosomes of different origins and enables efficient loading of different nucleic acid drugs on exosomes. Importantly, EAA-mediated loading of DMD exon-skipping PMO drugs on exosomes to form EXO$_{EAA-PMO}$ and systemic administration of EXO$_{EAA-PMO}$ at low doses elicited therapeutic levels of dystrophin restoration and significant functional improvements in *mdx* mice.

**Impact**

As a nucleic acid, EAA can be easily synthesized with nucleic acid therapeutics, enabling their efficient loading on exosomes without conjugation or modification orthogonal to CD63-binding peptide approach for peptide or protein loading, thus providing an easy and generalizable strategy for loading nucleic acid therapeutics on exosomes.

as follows: 3'-**CCGGTTT**GGAGCCGAAT**GGACTTTA**-5' (phosphorothioate ends highlighted in bold). The 5' and 3' ends of the probes were labeled with digoxigenin and biotin, respectively. Standard PMO samples and muscle tissues (100 mg/ml) were digested with 20 mg/ml proteinase K at 55 °C overnight. Following PMO-probe hybridization, the avidin-biotin interaction of the hybridized probe was performed on Pierce Neutr-Avidin Coated 96-well plates (Thermo Fisher Scientific, USA). Free probes were digested with 10 unit/μl of micrococcal nuclease (Thermo Fisher Scientific, USA). Then the hybridized probes were reacted with rabbit monoclonal antibody to digoxigenin (Cell Signalling Technology, USA), followed by detection with peroxidase-conjugated goat anti-rabbit IgG (Abcam, Cambridge, UK). Signals from the PMO-hybridized probe were detected at 450 nm with TME Substrate (Solarbio, Beijing, China) in a monochromator EnSpire Multimode plate reader (PerkinElmer, Boston, USA).

## Data analysis

All data are reported as mean values ± SEM. Statistical differences between different treated groups were evaluated by SigmaStat (Systat Software Inc. Chicago, IL, USA). Both parametric and nonparametric analyses were applied as specified in figure legends. Randomization was done for the animal experiments. The experiments were not blinded.

## Data availability

This study includes no data deposited in external repositories.

## Peer review information

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

## Acknowledgements

The authors thank Dr Weiting Wang (Tianjin Institute of Pharmaceutical Research, Tianjin, China) for the help with blood flow measurement and Professor Linli Lv (Institute of Nephrology, Zhongda Hospital, Southeast University School of Medicine, Nanjing, Jiangsu Province, China) for helping super-resolution microscopy, and Professor Ge Zhang (Law Sau Fai Institute for Advancing Translational Medicine in Bone and Joint Diseases, School of Chinese Medicine, Hong Kong Baptist University, Hong Kong SAR, China) for advice on the SELEX, and Core facility of Research Center of Basic Medical Sciences (Tianjin Medical University, Tianjin) for technical support, specifically the flow cytometry core facilities. This research was supported by National Natural Science Foundation of China (no. 82320108013, 82030054), Natural Science Foundation of Tianjin (no. 22JCYBJC00010), Open Competition to Select the Best Candidates" Key Technology Program for Nucleic Acid Drugs of NCTB (no. NCTIB2022HS01018) and Tianjin Municipal 13th five-year plan (Tianjin Medical University Talent Project).

## Author contributions

**Gang Han**: Conceptualization; Data curation; Software; Formal analysis; Funding acquisition; Validation; Visualization; Methodology; Writing—original draft; Project administration; Writing—review and editing. **Yao Zhang**: Data curation; Software; Methodology; Project administration. **Li Zhong**: Methodology; Project administration. **Biaobiao Wang**: Software; Methodology.

**Shuai Qiu**: Methodology. **Jun Song**: Methodology. **Caorui Lin**: Methodology. **Fangdi Zou**: Methodology. **Jingqiao Wu**: Methodology. **Huanan Yu**: Methodology. **Chao Liang**: Supervision. **Ke Wen**: Methodology. **Yiqi Seow**: Supervision; Writing—original draft; Writing—review and editing. **HaiFang Yin**: Conceptualization; Data curation; Software; Formal analysis; Supervision; Funding acquisition; Validation; Visualization; Writing—original draft; Project administration; Writing—review and editing.

## Disclosure and competing interests statement

The authors declare no competing interests.

# Expanded View Figures

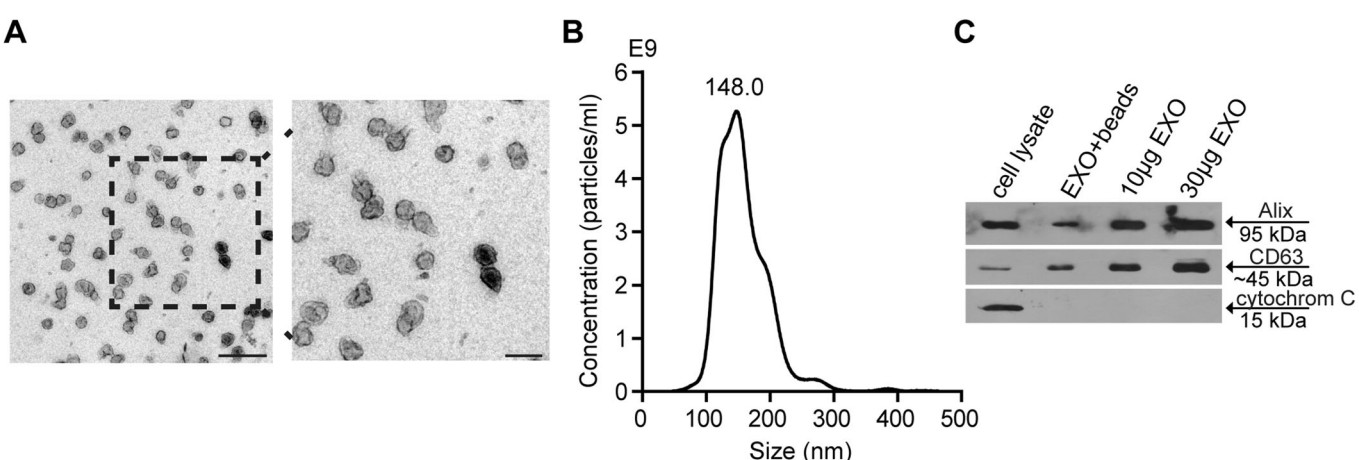

**Figure EV1. Characterization of murine myotube-derived exosomes and CP05-based binding assay.**

(A) Representative transmission electron microscopic (TEM) images of exosomes derived from murine myotubes (scale bar = 200 nm for right panel and 500 nm for the left). (B) Size distribution of exosomes derived from murine myotubes with nanoparticle tracking analysis (NTA). (C) Western blot to examine the binding of murine myotube-derived exosomes (EXO) to CP05-coated beads. The loading was specified in the figure.

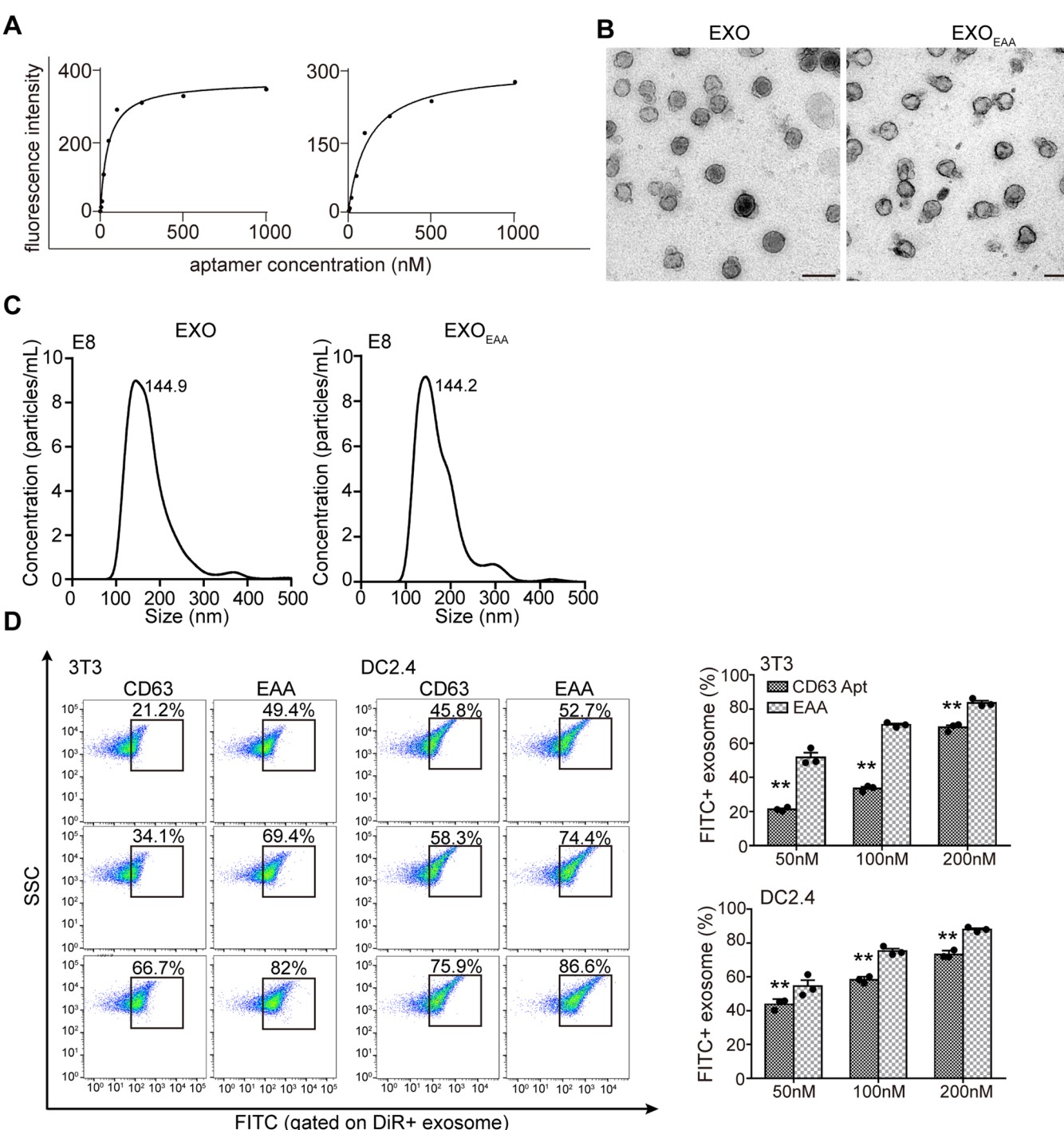

**Figure EV2. Characterization of murine myotube-derived exosomes modified with EAA and binding assay for EAA with different exosomes.**

(A) Measurement of binding affinity of EAA or CD63 aptamer to exosomes derived from mouse serum with flow cytometry. (B) Representative TEM images of murine myotube-derived exosomes modified with EAA (EXO$_{EAA}$) (scale bar = 200 nm). (C) Size distribution of EXO and EXO$_{EAA}$ with nanoparticle tracking analysis (NTA). (D) Flow cytometric and quantitative analysis of binding efficiency of EAA or CD63 aptamer to exosomes across different concentrations ($n = 3$). Exosomes were derived from murine 3T3 and DC2.4 cells. Data information: Data represent different numbers ($n$) of biological replicates. The data with error bars are shown as mean ± SEM. (D) Statistical significance was determined using two-tailed $t$ test. **$P < 0.001$.

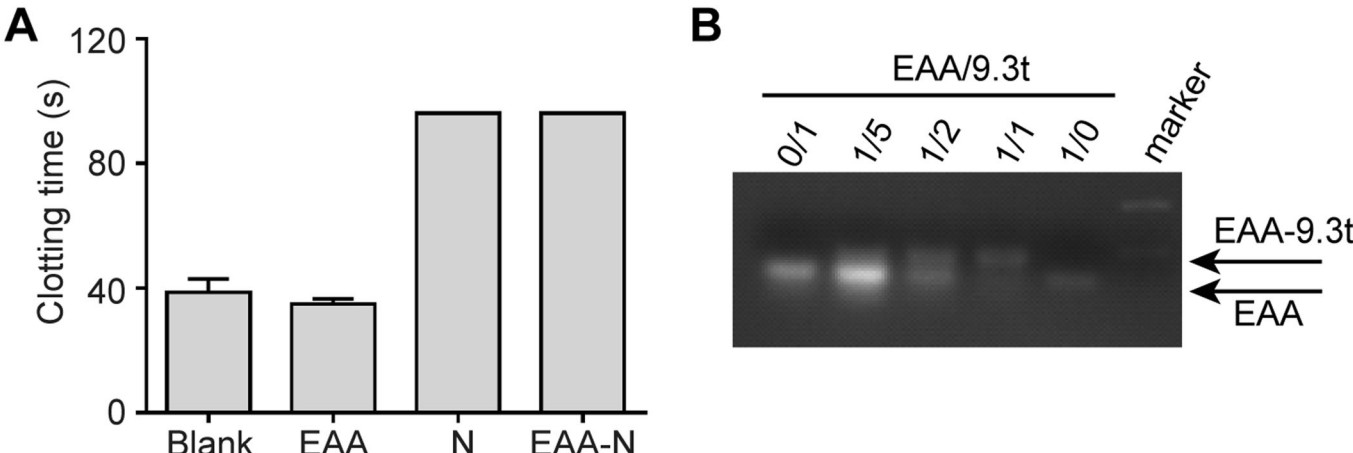

**Figure EV3.  In vitro evaluation of EAA-NU172's activity and optimization of binding ratio of EAA to 9.3t RNA aptamers.**

(A) Measurement of the clotting time of rat plasma after the addition of thrombin in the presence of EAA, NU172 (N)  or EAA-NU172 (EAA-N) ($n = 3$). Data represent different numbers ($n$) of biological replicates. The data with error bars are shown as mean ± SEM. (B) Agarose gel for mobility shift assay to determine the binding of EAA-linker to linker-9.3t at different molar ratios.

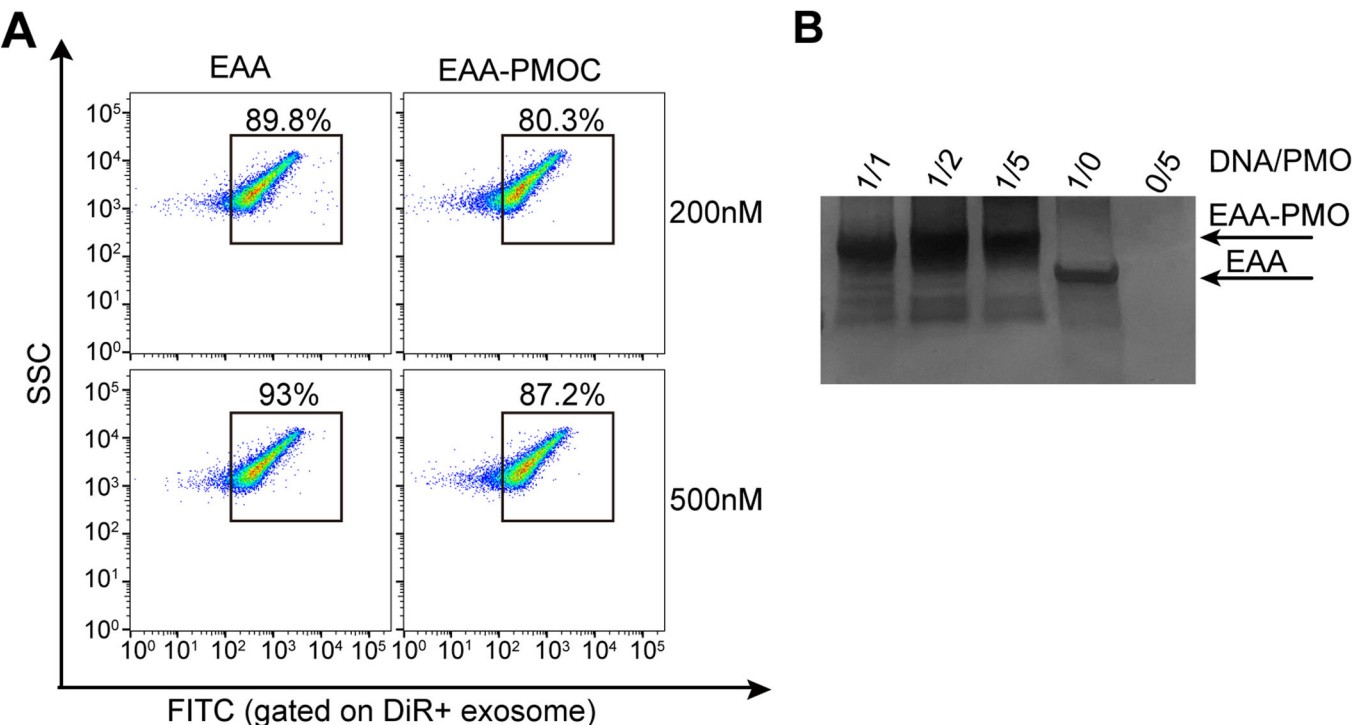

**Figure EV4. Evaluation of binding efficiency of EAA-PMOC to exosomes and optimization of binding of EAA-PMOC to PMO.**

(A) Flow cytometric analysis of binding efficiency of EAA-PMOC to exosomes derived from murine myotubes at two different concentrations. EAA-PMOC refers to EAA fused with a PMO- Complementary sequence. Aptamers were labeled with FITC and exosomes were labeled with DiR. (B) Silver staining for mobility shift assay to determine the binding of EAA-PMOC to PMO at different molar ratios.

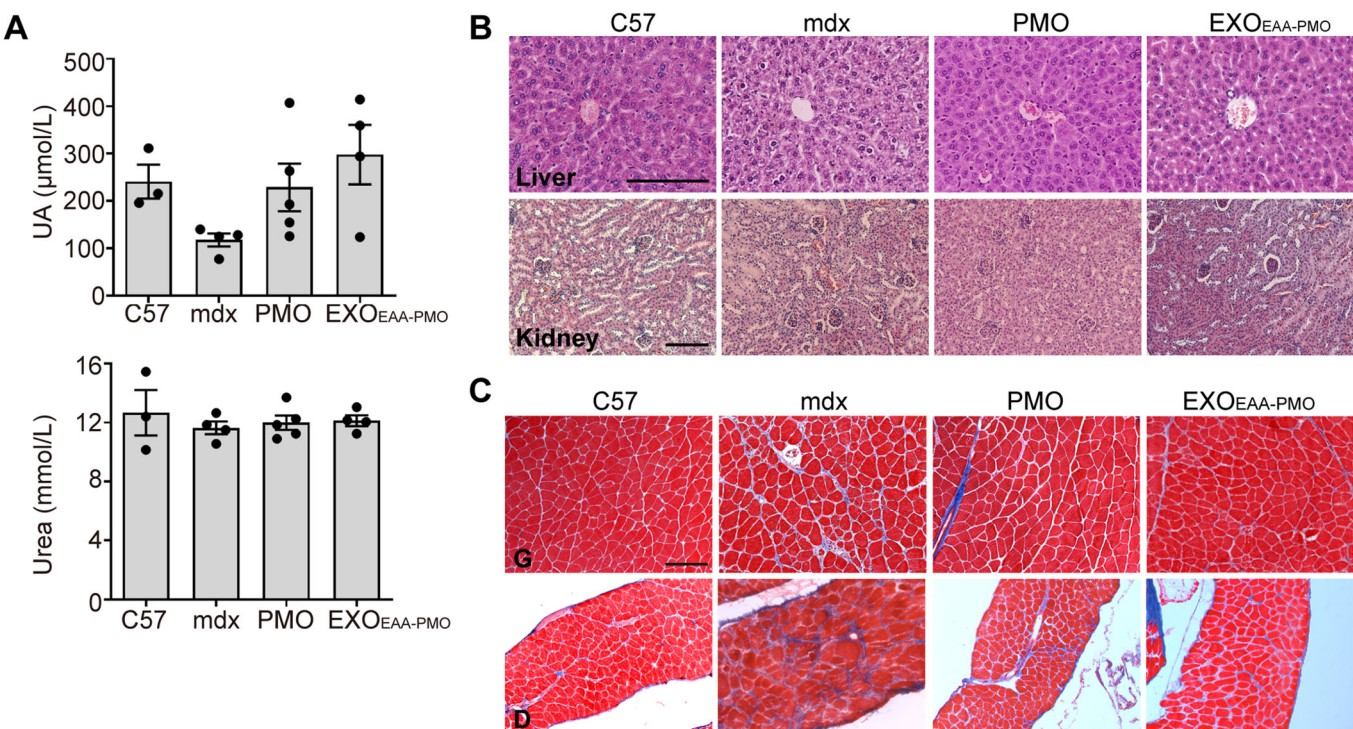

**Figure EV5.  Biochemical and histological examination of *mdx* mice treated with EXO<sub>EAA-PMO</sub>.**

EXO<sub>EAA-PMO</sub> at the PMO dose of 25 mg/kg were administered into adult *mdx* mice for three times weekly intravenously and tissues were harvested 2 weeks after last injection. (**A**) Analysis of biochemical indicators for kidney function in mdx mice treated with PMO (n = 5), EXO<sub>EAA-PMO</sub> (n = 4), untreated *mdx* (n = 4) and *C57BL/6* controls (n = 3). (**B**) H&E staining of liver and kidney from *mdx* mice treated with EXO<sub>EAA-PMO</sub> (scale bar = 100 μm). (**C**) Collagen deposition analysis in gastrocnemius (G) and quadriceps (Q) from treated *mdx* mice (scale bar = 100 μm). Data information: In (**A**), the data represent different numbers (*n*) of biological replicates. The data with error bars are shown as mean ± SEM.

