## [Peer Review File · EMBO Molecular Medicine]

Generalizable anchor aptamer strategy for loading nucleic acid therapeutics on exosomes

Gang Han, Yao Zhang, Li Zhong, Biaobiao Wang, Shuai Qiu, Jun Song, Caorui Lin, Fangdi Zou, Jingqiao Wu, Huanan Yu, Chao Liang, Ke Wen, Yiqi Seow, and Haifang Yin

Corresponding author: Haifang Yin (haifangyin@tmu.edu.cn)

Review Timeline:

Submission Date:	30th Nov 23
Editorial Decision:	22nd Dec 23
Revision Received:	29th Jan 24
Editorial Decision:	9th Feb 24
Revision Received:	15th Feb 24
Accepted:	19th Feb 24

Editor: Zeljko Durdevic

Transaction Report:

22nd Dec 2023

Dear Prof. Yin,

Thank you for the submission of your manuscript to EMBO Molecular Medicine. We have now received feedback from the two reviewers who agreed to evaluate your manuscript. Both referees recognize potential interest of the study but also raise important criticism that should be addressed in a major revision. Particular attention should be given to identifying the target of the aptamer and providing detailed information about the methodology especially about the number of biological replicates as suggested by the referee #2. If you would like to discuss further the points raised by the referees, I am available to do so via email or video. Let me know if you are interested in this option.

We would welcome the submission of a revised version within three months for further consideration. Please let us know if you require longer to complete the revision.

I look forward to receiving your revised manuscript.

Yours sincerely,

Zeljko Durdevic

We require:

- 1) A .docx formatted version of the manuscript text (including legends for main figures, EV figures and tables). Please make sure that the changes are highlighted to be clearly visible.
- 2) Individual production quality figure files as .eps, .tif, .jpg (one file per figure). For guidance, download the 'Figure Guide PDF': (<https://www.embopress.org/page/journal/17574684/authorguide#figureformat>).
- 3) A .docx formatted letter INCLUDING the reviewers' reports and your detailed point-by-point responses to their comments. As part of the EMBO Press transparent editorial process, the point-by-point response is part of the Review Process File (RPF), which will be published alongside your paper.
- 4) A complete author checklist, which you can download from our author guidelines (<https://www.embopress.org/page/journal/17574684/authorguide#submissionofrevisions>). Please insert information in the checklist that is also reflected in the manuscript. The completed author checklist will also be part of the RPF.
- 5) Please note that all corresponding authors are required to supply an ORCID ID for their name upon submission of a revised manuscript.

6) It is mandatory to include a 'Data Availability' section after the Materials and Methods. Before submitting your revision, primary datasets produced in this study need to be deposited in an appropriate public database, and the accession numbers and database listed under 'Data Availability'. Please remember to provide a reviewer password if the datasets are not yet public (see <https://www.embopress.org/page/journal/17574684/authorguide#dataavailability>).

13) Author contributions: You will be asked to provide CRediT (Contributor Role Taxonomy) terms in the submission system. These replace a narrative author contribution section in the manuscript.

14) A Conflict of Interest statement should be provided in the main text.

Please also suggest a striking image or visual abstract to illustrate your article as a PNG file 550 px wide x 300-800 px high.

**** Reviewer's comments ****

Referee #1 (Comments on Novelty/Model System for Author):

Despite the ongoing gene therapy for DMD, exon skipping remains as a promising approach. The biggest hurdle for high efficacy of ASO treatment is the low delivery efficiency. It is a big reap forward when such a low dose of PMO can achieve the reported efficacy in vivo systemically.

Referee #1 (Remarks for Author):

This manuscript by Gang Han et al titled "Generalizable anchor aptamer strategy for loading 1 nucleic acid therapeutics on exosomes" identified an myotube-derived exosome-binding DNA aptamer (Exosomal Anchor Aptamer-EAA) with high binding affinity to exosomes of different origins, and enables efficient loading of different nucleic acid drugs on exosomes. The authors show the prolonged serum stability of thrombin DNA aptamer inhibitor Nu172 by exosome-loading, resulting in increased blood flow after injury in vivo. The authors also show that phosphorodiamidate morpholino oligomers (PMOs) can be readily loaded on exosomes by incorporating a complementary sequence to EAA (EXO-EAA-PMO) eliciting significantly greater muscle cell uptake, tissue accumulation, and targeted exon skipping and dystrophin expression in vitro and in vivo. Systemic administration of EXO-EAA-PMO at the PMO dose of 25mg/kg/ weekly for 3 weeks elicited therapeutic levels of dystrophin restoration and significant functional improvements without detectable adverse effect in mdx mice, indicating the potency of exosomes as viable delivery vehicles for PMOs. The results are very interesting as such a low dose of PMO can achieve apparently about 5-10% dystrophin restoration. Importantly the same approach can be used to identify and deliver many other therapeutic oligonucleotides.

There are some concerns:

1. Authors need to discuss how the use of exosome can be realized for clinical application to DMD, which requires life-long treatment.
2. It is also important to discuss the possible reasons for lack of any delivery of exosome-carried cargo to cardiac muscle.

Some sentence may require careful reorganization to be concise and easy to understand by readers. For example, "...the DNA aptamer targeting CD63, an exosomal biomarker transmembrane protein". Here the word "biomarker" between "exosomal" and "transmembrane protein" only confuse readers. Either "a protein biomarker of exosome" or an "a biomarker of exosome" should be sufficient.

Pag 171-173:

".... FACS showed an equilibrium dissociation constant (KD) value of 51.38{plus minus}5.12nM, which shows a stronger binding affinity than CD63 DNA aptamer...". The underlined should be removed.

Referee #2 (Comments on Novelty/Model System for Author):

The technical quality is somewhat hard to assess as the authors makes no mention whatsoever of what the target for their aptamer could be, and seems not to have investigated it. Also, this reviewer is not skilled in flow cytometry, and can not reasonable assess e.g. the gating strategies. More replicate animals in the in vivo experiments would have been desirable.

Novelty: The whole-exosome-binding with aptamers is reasonably novel.

Medical impact: This is very early stage, and the path to medical PoC in humans is still quite long.

The model systems are OK.

Referee #2 (Remarks for Author):

This work is interesting, and its use for drug delivery holds some promise.

One major shortcoming stands out: The authors make no experimental (or even theoretical) attempt to identify the molecular binding target of their aptamer. This should be done, and would be quite transformative if experimentally proven.

Other comments:

1. Latter part of Introduction (line 118 onwards) is really a results summary and excessively long as compared to the rest of the Intro.
2. It would be very interesting to see a screen of a wide range of exosomes from diverse sources, to justify the claim 'irrespective of sources. (line 186).
3. The animal experiments seems to have been done with no replicates? Replicates would improve the work significantly.
4. It is assumed that equimolar amounts of free and complexed PMOs are dosed, but it is not said explicitly. This needs clarification in the text.

- Particular attention should be given to identifying the target of the aptamer and providing detailed information about the methodology especially about the number of biological replicates as suggested by the referee #2.

Response: We thank the editor for highlights and we have addressed these concerns with great details as stated in **Point 1, 4 and 7** (Referee#2). Also we have added more details especially for the number of biological replicates in the Materials and Methods section as the editor has recommended.

Referee 1

Point 1: Authors need to discuss how the use of exosome can be realized for clinical application to DMD, which requires life-long treatment.

Response: We thank Referee#1 for the helpful suggestion and have discussed this point in Discussion as Referee#1 has recommended. Now it reads: "Exosomes have been extensively tested in clinical trials for a myriad of diseases (Tan, Li et al., 2024), and also being actively exploited for treating DMD either as therapeutics or delivery vehicles (Yedigaryan & Sampaolesi, 2023). Although most studies demonstrated therapeutic effects on skeletal muscles with unmodified or modified exosomes from different sources (Gao et al., 2018, Leng, Dong et al., 2021, Ran, Gao et al., 2020, Sandona, Consalvi et al., 2020), Marban and colleagues reported that exosomes from clinical stage cardiosphere-derived cells improved both cardiac and skeletal

muscle functions in DMD models (Aminzadeh, Rogers et al., 2018, Rogers, Fournier et al., 2019), showing the potential and clinical applicability of exosomes for treating DMD. Importantly, it seems that the source of exosomes might have impact on tissue distribution, with exosomes from cardiosphere-derived cells showing superior heart- targeting property to exosomes from other sources. We speculated that cardiogenic moieties on exosomes from cardiosphere-derived cells might be primarily responsible for targeting the heart though other possibilities cannot be excluded. Functional and tissue-targeted modification of exosomes from clinical stage cells such as cardiosphere-derived cells (in Phase III clinical trial NCT05126758) or other stem cells would further increase efficacy in target tissues. Worth mentioning, PMOs are expensive to manufacture and even more expensive to conjugate. If there were methods to increase efficacy per PMO molecule injected, it could make a big difference in the affordability of the approach. Exosomes can extend circulatory half- life of PMO tremendously, lowering required dosage, but if loading onto exosomes and targeting exosomes were prohibitively expensive, that would defeat any savings arising from reduced dosage. Previously, we had identified CP05 which was a peptide that can be used to modify exosome surface (Gao et al., 2018), but conjugating peptides to nucleic acids is still more expensive than lengthening the synthesis length of an oligonucleotide, thus EAA is a new tool in the arsenal to help enhance exosomes to a point where PMOs could be dosed at very low doses, enabling greater access to therapy for DMD patients, however often they need to take the drugs. Also exosomes were shown to be able to deliver Cas9 ribonucleoprotein complexes to liver (Wan, Zhong et al., 2022) and repeated administration might not be necessary for DMD patients in future if Cas9 and single guide (sg)RNA can be co-delivered to muscle and heart efficiently. Thus, new approaches for efficient loading of nucleic acids to exosomes will accelerate their clinical use.”

Point 2: It is also important to discuss the possible reasons for lack of any delivery of exosome-carried cargo to cardiac muscle.

Response: We are grateful for Referee#1’s comments and have discussed this point in Discussion as Refereer#1 has recommended. Now it reads: “Exosomes have been extensively tested in clinical trials for a myriad of diseases (Tan, Li et al., 2024), and also being actively exploited for treating DMD either as therapeutics or delivery vehicles (Yedigaryan & Sampaolesi, 2023). Although most studies demonstrated therapeutic effects on skeletal muscles with unmodified or modified exosomes from different sources (Gao et al., 2018, Leng, Dong et al., 2021, Ran, Gao et al., 2020, Sandona, Consalvi et al., 2020), Marban and colleagues reported that exosomes from clinical stage cardiosphere-derived cells improved both cardiac and skeletal muscle functions in DMD models (Aminzadeh, Rogers et al., 2018, Rogers, Fournier et al., 2019), showing the potential and clinical applicability of exosomes for treating DMD. Importantly, it seems that the source of exosomes might have impact on tissue distribution, with exosomes from cardiosphere-derived cells showing superior heart- targeting property to exosomes from other sources. We speculated that cardiogenic moieties on exosomes from cardiosphere-derived cells might be primarily responsible for targeting the heart though other possibilities cannot be excluded. Functional and tissue-targeted modification of exosomes from clinical stage cells such as cardiosphere-derived cells (in Phase III clinical trial NCT05126758) or other stem cells would further increase efficacy in target tissues. Worth mentioning, PMOs are expensive to manufacture and even more expensive to conjugate. If there were methods to increase efficacy per PMO molecule injected, it could make a big difference in the affordability of the approach. Exosomes can extend circulatory half- life of PMO tremendously, lowering required dosage, but if loading onto exosomes and targeting exosomes were prohibitively

expensive, that would defeat any savings arising from reduced dosage. Previously, we had identified CP05 which was a peptide that can be used to modify exosome surface (Gao et al., 2018), but conjugating peptides to nucleic acids is still more expensive than lengthening the synthesis length of an oligonucleotide, thus EAA is a new tool in the arsenal to help enhance exosomes to a point where PMOs could be dosed at very low doses, enabling greater access to therapy for DMD patients, however often they need to take the drugs. Also exosomes were shown to be able to deliver Cas9 ribonucleoprotein complexes to liver (Wan, Zhong et al., 2022) and repeated administration might not be necessary for DMD patients in future if Cas9 and single guide (sg)RNA can be co-delivered to muscle and heart efficiently. Thus, new approaches for efficient loading of nucleic acids to exosomes will accelerate their clinical use.”

Point 3: Some sentence may require careful reorganization to be concise and easy to understand by readers. For example, " ...the DNA aptamer targeting CD63, an exosomal biomarker transmembrane protein". Here the word "biomarker" between "exosomal" and transmembrane protein" only confuse readers. Either "a protein biomarker of exosome" or an "a biomarker of exosome" should be sufficient.

Response: We thank Referee#1 for the kind reminder and we have revised the text as Referee#1 has suggested. Also we have checked carefully throughout to make sure that the sentences are easier to understand.

Point 4: Pag 171-173:

"... FACS showed an equilibrium dissociation constant (KD) value of 51.38{plus minus}5.12nM, which shows a stronger binding affinity than CD63 DNA aptamer...". The underlined should be removed.

Response: We thank Referee#1 for the comment, but the underlined mark wasn't retained in the referees' comment we received. We would be happy to edit if Referee#1 can send it with different annotations.

Referee 2

Point 1: The technical quality is somewhat hard to assess as the authors makes no mention whatsoever of what the target for their aptamer could be, and seems not to have investigated it.

Response: We agree with Referee#2 that we have not yet to be able to identify the target of the aptamer. As we described in the manuscript, EAA was identified by panning against the entire exosome rather than any specific target, to give us a chance to identify surface features that may not be previously linked to exosomes. While we did not present all the results as some were negative, there were several attempts to characterize and identify its potential targets. 1) To determine whether proteins are the potential targets, we digested proteins on the surface of exosomes with trypsin and the results showed that digestion of proteins on the surface of exosome significantly compromised the binding of EAA (Figure 1a, shown below). 2) To investigate whether the most common protein biomarkers on the surface of exosome including CD63, CD9 and CD81 are the potential targets, we neutralized CD63, CD9 or CD81 with corresponding antibodies. However, neither blocking CD63, CD9 or CD81 individually nor in combination reduced the binding of EAA to exosomes (Figure 1b, shown below). Furthermore, we tried to pull-down the binding partners of EAA via co-immunoprecipitation with biotin-tagged EAA. Although a few bands appeared in the co-immunoprecipitation experiment, mass spectrometry has not yielded any plausible candidate. While specific targets, such as CD63, have

been previously targeted, we hypothesized that the surface proteome of the exosomes may harbour more features that can be used to anchor targeting or therapeutic molecules beyond the known proteins. This is akin to performing phage display on whole organs rather than a protein target, which is a well-established panning approach. There are only a very limited number of tools available for exosome binding. We believe new approaches need to be developed to complement traditional approaches for identification of binding partners for aptamers against exosome, a nano-scaled target with high complexity. Importantly, the primary objective of our current study was to demonstrate that EAA enables facile and efficient loading of nucleic acid drugs of different chemical backbones on exosomes, thus providing a generalizable strategy for nucleic acid drug loading, potentially enabling a route towards more effective and affordable exosome-based oligonucleotide therapies. This is a timely report and urgently needed given the rapid growth of nucleic acid therapeutics. And we would like to report the whole set-up for receptor identification of EAA prior to clinical deployment separately in due course. Nevertheless, we have extensively discussed this point in Discussion as Referee#2 has recommended. Now it reads: “Compared to the CD63-binding aptamer, exosome binding affinity and copies per exosome is significantly higher for EAA. We speculated that it is likely due to the abundance of its targets on exosomes as EAA was identified against intact exosomes rather than a single protein as CD63 DNA aptamer (Zhou, Rahimian et al., 2016). Digestion of proteins on the surface of exosomes with trypsin significantly compromised the binding of EAA but not by blocking common protein biomarkers on the surface of exosomes including CD63, CD9 or CD81 individually or in combination with antibodies (data not shown). These findings suggest that EAA specifically binds to protein on the surface of exosomes but not to CD63, CD9 or CD81. However, identification of binding partners for EAA is challenging due to the highly complex surface molecules of exosomes. More extensive studies are warranted to identify the receptor(s) of EAA in future.”

a

b

Figure 1. Flow cytometric analysis of binding efficiency of EAA to exosomes under different conditions. Exosomes were derived from differentiated C2C12 cells and EAA was labeled with Cy5. (a) Flow cytometric analysis of binding efficiency of EAA to exosomes after digestion with trypsin, a protocol adopted from a previous study (Wang C., et al. *Journal of Extracellular Vesicles* (2020) 9:1746529). (b) Flow cytometric analysis of binding efficiency of EAA to exosomes blocked with CD63, CD9 or CD81 antibodies individually or in combination.

Point 2: Also, this reviewer is not skilled in flow cytometry, and can not reasonable assess e.g. the gating strategies.

Response: We are grateful for Referee#2's comments and have provided detailed gating strategy in Appendix Figures (Appendix Figure S2) as Referee#2 has suggested. Briefly, exosomes were gated as described previously (Gao, et al. *Science Translational Medicine* (2018)10:eaat0195), followed by gating with DiR-positive exosomes and then DiR- and FITC-positive exosomes, in which EAA was labeled with FITC.

Point 3: More replicate animals in the in vivo experiments would have been desirable.

Response: We thank Referee#2 for the comment. To be clear, we conducted all the animal experiments with strict compliance with 3Rs principle (Replacement, Reduction and Refinement) as required for animal welfare and aimed to achieve valid conclusions with a minimal number of animals used. Based on this, we performed two different animal experiments (arterial thrombosis and DMD) in our present study. For the arterial thrombosis experiment, real-time imaging of blood flow with small animal laser Doppler (Figure 3E) was shown with four animals per group as specified in Figure legend. However, we do apologize for the missing details for the quantitative analysis of arterial blood flow with blood flow measurement devices (Figure 3F), in which "mice were treated with NU172 (N) (n=5), the mixture of NU172 and exosomes (EXO/N) (n=5), EXO_{EAA-N} (n=10), untreated (n=5) or sham controls (n=6). Blank refers to untreated controls" and we have added these information in the Figure legend (original data were supplied in Source Data). The data were presented as mean±SEM, which showed dramatic difference between treated and untreated groups (e.g. with or without blood flow) with small variations between biological replicates, adequately supporting the conclusion that EXO_{EAA-N} significantly increased the blood flow compared to other groups. For the DMD experiment, the *mdx* mouse was used, which is a transgenic DMD mouse model with spontaneous mutation in exon23 of murine *dystrophin* gene and the most commonly used animal model for DMD (McGreevy et al. *Dis. Model Mech.* (2015) 8(3): 195-213). For the local intramuscular study, immunostaining and quantitative analysis of dystrophin-positive (dys⁺) myofibres in tibialis anterior (TA) muscles of *mdx* mice treated with PMO or EXO_{EAA-PMO} were shown (Figure 4D), in which five mice per group were used as specified in Figure legend. For the systemic intravenous study, muscle function (Figure 5C), measurement of serum creatine kinase (CK) levels (Figure 5D) and analysis of serum levels of liver enzymes (Figure 5F) in *mdx* mice treated with PMO (n=5) or EXO_{EAA-PMO} (n=4), untreated *mdx* (n=4) and wild-type controls (n=3) were shown. For the measurement of PMOs in muscle tissues from treated *mdx* mice, four mice per group were tested as specified in Figure legend (Figure 5E). Although different numbers of animal have been used in previous studies for DMD exon-skipping therapeutics ranging from three to six or more (e.g. Alter et al. *Nat. Med.* (2006) 12(2): 175-177; Lu et al. *PNAS* (2004) 102(1):198-203; two studies that are cornerstones for DMD therapeutics), the validity of conclusion matters the most. In our DMD experiment, significant difference was achieved between EXO_{EAA-PMO} and PMO with the number of animals used, supporting the conclusion that exosomes promote the delivery of PMO

to muscle. Surely, we agree with Referee#2 that a larger dataset will be needed to advance this therapy to human use, but as this is not the final iteration of the design, we will further improve the therapy in subsequent publications. Using the principle of the 3Rs, we strive to use the minimal number of animals to achieve a statistically significant and meaningful dataset; hence we hope Referee#2 can accept our rationale in light of these principles. Also we have added more details regarding the number of animals in Materials and Methods as Referee#2 and the editor have recommended.

Point 4: The authors make no experimental (or even theoretical) attempt to identify the molecular binding target of their aptamer. This should be done, and would be quite transformative if experimentally proven.

Response: As stated in **Point 1** (Referee#2), we thank Referee#2 for this comment. As we described in the manuscript, EAA was identified by panning against the entire exosome rather than any specific target, to give us a chance to identify surface features that may not be previously linked to exosomes. While we did not present all the results as some were negative, there were several attempts to characterize and identify its potential targets. **1)** To determine whether proteins are the potential targets, we digested proteins on the surface of exosomes with trypsin and the results showed that digestion of proteins on the surface of exosome significantly compromised the binding of EAA (Figure 1a, shown above). **2)** To investigate whether the most common protein biomarkers on the surface of exosome including CD63, CD9 and CD81 are the potential targets, we neutralized CD63, CD9 or CD81 with corresponding antibodies. However, neither blocking CD63, CD9 or CD81 individually nor in combination reduced the binding of EAA to exosomes (Figure 1b, shown above). Furthermore, we tried to pull-down the binding partners of EAA via co-immunoprecipitation with biotin-tagged EAA. Although a few bands appeared in the co-immunoprecipitation experiment, mass spectrometry has not yielded any plausible candidate. While specific targets, such as CD63, have been previously targeted, we hypothesized that the surface proteome of the exosomes may harbour more features that can be used to anchor targeting or therapeutic molecules beyond the known proteins. This is akin to performing phage display on whole organs rather than a protein target, which is a well-established panning approach. There are only a very limited number of tools available for exosome binding. We believe new approaches need to be developed to complement traditional approaches for identification of binding partners for aptamers against exosome, a nano-scaled target with high complexity. Importantly, the primary objective of our current study was to demonstrate that EAA enables facile and efficient loading of nucleic acid drugs of different chemical backbones on exosomes, thus providing a generalizable strategy for nucleic acid drug loading, potentially enabling a route towards more effective and affordable exosome-based oligonucleotide therapies. This is a timely report and urgently needed given the rapid growth of nucleic acid therapeutics. And we would like to report the whole set-up for receptor identification of EAA prior to clinical deployment separately in due course. Nevertheless, we have extensively discussed this point in Discussion as Referee#2 has recommended. Now it reads: "Compared to the CD63-binding aptamer, exosome binding affinity and copies per exosome is significantly higher for EAA. We speculated that it is likely due to the abundance of its targets on exosomes as EAA was identified against intact exosomes rather than a single protein as CD63 DNA aptamer (Zhou, Rahimian et al., 2016). Digestion of proteins on the surface of exosomes with trypsin significantly compromised the binding of EAA but not by blocking common protein biomarkers on the surface of exosomes including CD63, CD9 or CD81 individually or in combination with antibodies (data not shown). These findings suggest

that EAA specifically binds to protein on the surface of exosomes but not to CD63, CD9 or CD81. However, identification of binding partners for EAA is challenging due to the highly complex surface molecules of exosomes. More extensive studies are warranted to identify the receptor(s) of EAA in future.”

Point 5: Latter part of Introduction (line 118 onwards) is really a results summary and excessively long as compared to the rest of the Intro.

Response: We thank Referee#2 for the comments and have reorganized the last part of Introduction as Referee#2 has recommended.

Point 6: It would be very interesting to see a screen of a wide range of exosomes from diverse sources, to justify the claim 'irrespective of sources. (line 186).

Response: We thank Referee#2 for the suggestion. In the present study, we tested exosomes from human 293T cells, murine 3T3 and DC2.4 cells and rat serum. To expand further, we have included exosomes from human mesenchymal stem cells and mouse serum (Figure 2D and 2E) as Referee#2 has recommended. Also, we have included the binding affinity data for mouse serum in the manuscript as Expanded View Figure (Figure EV3A).

Point 7: The animal experiments seems to have been done with no replicates? Replicates would improve the work significantly.

Response: As stated in **Point 3** (Referee#2), we conducted all the animal experiments with strict compliance with 3Rs principle (Replacement, Reduction and Refinement) as required for animal welfare and aimed to achieve valid conclusions with a minimal number of animals used. Based on this, we performed two different animal experiments (arterial thrombosis and DMD) in our present study. For the arterial thrombosis experiment, real-time imaging of blood flow with small animal laser Doppler (Figure 3E) was shown with four animals per group as specified in Figure legend. However, we do apologize for the missing details for the quantitative analysis of arterial blood flow with blood flow measurement devices (Figure 3F), in which “mice were treated with NU172 (N) (n=5), the mixture of NU172 and exosomes (EXO/N) (n=5), EXO_{EAA-N} (n=10), untreated (n=5) or sham controls (n=6). Blank refers to untreated controls” and we have added these information in the Figure legend (original data were supplied in Source Data). The data were presented as mean±SEM, which showed dramatic difference between treated and untreated groups (e.g. with or without blood flow) with small variations between biological replicates, adequately supporting the conclusion that EXO_{EAA-N} significantly increased the blood flow compared to other groups. For the DMD experiment, the *mdx* mouse was used, which is a transgenic DMD mouse model with spontaneous mutation in exon23 of *dystrophin* gene and the most commonly used animal model for DMD (McGreevy et al. *Dis. Model Mech.* (2015) 8(3): 195-213). For the local intramuscular study, immunostaining and quantitative analysis of dystrophin-positive (dys⁺) myofibres in tibialis anterior (TA) muscles of *mdx* mice treated with PMO or EXO_{EAA-PMO} were shown (Figure 4D), in which five mice per group were used as specified in Figure legend. For the systemic intravenous study, muscle function (Figure 5C), measurement of serum creatine kinase (CK) levels (Figure 5D) and analysis of serum levels of liver enzymes (Figure 5F) in *mdx* mice treated with PMO (n=5) or EXO_{EAA-PMO} (n=4), untreated *mdx* (n=4) and wild-type controls (n=3) were shown. For the measurement of PMOs in muscle tissues from treated *mdx* mice, four mice per group were tested as specified in Figure legend (Figure 5E). Although different numbers of animal have been used in previous studies for DMD exon-skipping therapeutics ranging from three to six or more (e.g. Alter et al. *Nat. Med.* (2006)

12(2): 175-177; Lu et al. *PNAS* (2004) 102(1):198-203; two studies that are cornerstones for DMD therapeutics), the validity of conclusion matters the most. In our DMD experiment, significant difference was achieved between EXO_{EAA-PMO} and PMO with the number of animals used, supporting the conclusion that exosomes promote the delivery of PMO to muscle. Surely, we agree with Referee#2 that a larger dataset will be needed to advance this therapy to human use, but as this is not the final iteration of the design, we will further improve the therapy in subsequent publications. Using the principle of the 3Rs, we strive to use the minimal number of animals to achieve a statistically significant and meaningful dataset; hence we hope Referee#2 can accept our rationale in light of these principles. Also we have added more details regarding the number of animals in Materials and Methods as Referee#2 and the editor have recommended.

Point 8: It is assumed that equimolar amounts of free and complexed PMOs are dosed, but it is not said explicitly. This needs clarification in the text.

Response: Yes, equimolar PMOs were used for free PMO and EXO_{EAA-PMO} in local and systemic studies for DMD. We have added more details in Materials and Methods as Referee#2 has recommended.

We very much hope that you will find our revised manuscript and detailed responses to the reviewers' comments satisfactory, and will now consider the manuscript suitable for publication in *EMBO Molecular Medicine*. We believe our manuscript to be of importance, principally as it is the first proof-of-concept study to demonstrate that EAA enables facile and efficient loading of various nucleic acid drugs on exosomes, thus providing a generalizable strategy for loading nucleic acid drugs on exosomes, potentially enabling a route towards more effective and affordable exosome-based oligonucleotide therapies.

We look forward to hearing from you in due course.

With best wishes

HaiFang Yin

9th Feb 2024

Dear Prof. Yin,

Thank you for the submission of your revised manuscript to EMBO Molecular Medicine. I am pleased to inform you that we will be able to accept your manuscript pending the following final amendments:

1) In the main manuscript file, please do the following:

- Please address all comments suggested by our data editors listed below:

o Figure legends:

1. Please note that a separate 'Data Information' section is required in the legends of figures 2c, e; 3b, d, h; 5c-f.

2. Please note that information related to n is missing in the legends of figure 3c; EV 3a.

3. Although 'n' is provided, please describe the nature of entity for 'n' in the legends of figures 2c, e; 3b, d, h; EV 2d.

4. Please note that the error bars are not defined in the legends of figures 2e; 3b-d, f, h; 4d; 5c-f; EV 2d; EV 3a; EV 5a.

- Correct callout of 3H to Figure 3H.

- Remove data not shown (p. 16).

- Rename "Conflict of Interest" to "Disclosure Statement & Competing Interests". We updated our journal's competing interests policy in January 2022 and request authors to consider both actual and perceived competing interests. Please review the policy <https://www.embopress.org/competing-interests> and update your competing interests if necessary.

- Author contributions: Please remove it from the manuscript and specify author contributions in our submission system. CRediT has replaced the traditional author contributions section because it offers a systematic machine-readable author contributions format that allows for more effective research assessment. You are encouraged to use the free text boxes beneath each contributing author's name to add specific details on the author's contribution. More information is available in our guide to authors:

<https://www.embopress.org/page/journal/17574684/authorguide#authorshipguidelines>

- Add data availability statement. This paragraph should contain information about deposited data produced in this study. If no data were deposited, please add the sentence: "This study includes no data deposited in external repositories".

- Correct the reference citation in the reference list. In the reference list, citations should be listed in alphabetical order. Where there are more than 10 authors on a paper, 10 will be listed, followed by "et al.". Please check "Author Guidelines" for more information.

<https://www.embopress.org/page/journal/17574684/authorguide#referencesformat>

2) Table: Please remove it from the main manuscript file, add it to the Appendix and rename it to Appendix Table S1. Also, update table callouts in the main text.

3) Appendix: Please upload it as a pdf file and correct Appendix S1 to Appendix Figure S1.

4) The Paper Explained: Please add it to the main manuscript text. Please check "Author Guidelines" for more information.

<https://www.embopress.org/page/journal/17574684/authorguide#researcharticleguide>

5) Synopsis: Please check your synopsis text and image before submission with your revised manuscript. Please be aware that in the proof stage minor corrections only are allowed (e.g., typos).

6) Source Data: Please upload one file per figure.

7) For more information: This space should be used to list relevant web links for further consultation by our readers. Could you identify some relevant ones and provide such information as well? Some examples are patient associations, relevant databases, OMIM/proteins/genes links, author's websites, etc...

8) As part of the EMBO Publications transparent editorial process initiative (see our Editorial at <http://embomolmed.embopress.org/content/2/9/329>), EMBO Molecular Medicine will publish online a Review Process File (RPF) to accompany accepted manuscripts. This file will be published in conjunction with your paper and will include the anonymous referee reports, your point-by-point response and all pertinent correspondence relating to the manuscript. Let us know whether you agree with the publication of the RPF and as here, if you want to remove or not any figures from it prior to publication. Please note that the Authors checklist will be published at the end of the RPF.

9) Please provide a point-by-point letter INCLUDING my comments as well as the reviewer's reports and your detailed responses (as Word file).

I look forward to reading a new revised version of your manuscript as soon as possible.

Yours sincerely,

Zeljko Durdevic

*** Instructions to submit your revised manuscript ***

- 1) a .docx formatted version of the manuscript text (including Figure legends and tables)
 - 2) Separate figure files*
 - 3) supplemental information as Expanded View and/or Appendix. Please carefully check the authors guidelines for formatting Expanded view and Appendix figures and tables at <https://www.embopress.org/page/journal/17574684/authorguide#expandedview>
 - 4) a letter INCLUDING the reviewer's reports and your detailed responses to their comments (as Word file).
 - 5) The paper explained: EMBO Molecular Medicine articles are accompanied by a summary of the articles to emphasize the major findings in the paper and their medical implications for the non-specialist reader. Please provide a draft summary of your article highlighting
 - the medical issue you are addressing,
 - the results obtained and
 - their clinical impact.This may be edited to ensure that readers understand the significance and context of the research. Please refer to any of our published articles for an example.
 - 6) For more information: There is space at the end of each article to list relevant web links for further consultation by our readers. Could you identify some relevant ones and provide such information as well? Some examples are patient associations, relevant databases, OMIM/proteins/genes links, author's websites, etc...
 - 7) Author contributions: the contribution of every author must be detailed in a separate section.
 - 8) EMBO Molecular Medicine now requires a complete author checklist (<https://www.embopress.org/page/journal/17574684/authorguide>) to be submitted with all revised manuscripts. Please use the checklist as guideline for the sort of information we need WITHIN the manuscript. The checklist should only be filled with page numbers where the information can be found. This is particularly important for animal reporting, antibody dilutions (missing) and exact values and n that should be indicated instead of a range.
 - 9) Every published paper now includes a 'Synopsis' to further enhance discoverability. Synopses are displayed on the journal webpage and are freely accessible to all readers. They include a short stand first (maximum of 300 characters, including space) as well as 2-5 one sentence bullet points that summarise the paper. Please write the bullet points to summarise the key NEW findings. They should be designed to be complementary to the abstract - i.e. not repeat the same text. We encourage inclusion of key acronyms and quantitative information (maximum of 30 words / bullet point). Please use the passive voice. Please attach these in a separate file or send them by email, we will incorporate them accordingly.
- You are also welcome to suggest a striking image or visual abstract to illustrate your article. If you do please provide a jpeg file 550 px-wide x 300-800px high.
- 10) A Conflict of Interest statement should be provided in the main text

11) Please note that we now mandate that all corresponding authors list an ORCID digital identifier. This takes <90 seconds to complete. We encourage all authors to supply an ORCID identifier, which will be linked to their name for unambiguous name identification.

Currently, our records indicate that the ORCID for your account is 0000-0001-8774-2864.

Link Not Available

Photos 400-800 DPI

*Additional important information regarding figures and illustrations can be found at

<https://bit.ly/EMBOPressFigurePreparationGuideline>. See also figure legend preparation guidelines:

<https://www.embopress.org/page/journal/17574684/authorguide#figureformat>

***** Reviewer's comments *****

Referee #2 (Remarks for Author):

Although it would still be nice to have data on the binding target, the manuscript has improved during review and is valuable for the scientific community

The authors addressed the remaining editorial issues.

19th Feb 2024

Dear Prof. Yin,

We are pleased to inform you that your manuscript is accepted for publication and is now being sent to our publisher to be included in the next available issue of EMBO Molecular Medicine.

Yours sincerely,
